# EXPLAINING GROKKING THROUGH CIRCUIT EFFICIENCY

## ABSTRACT

We present a theory of grokking in neural networks which explains grokking in terms of the relative efficiency of competing emergent sub-networks (circuits). Grokking is an important generalisation phenomenon where continuing to train a network which already achieves nearly perfect training loss can still dramatically improve the test loss. Our theory explains why generalising circuits gradually outcompete memorising circuits. This is because memorising circuits are inefficient for compressing large datasets—the per-example cost is high—while generalising circuits have a larger fixed cost but better per-example efficiency. Strikingly, our theory is precise enough to produce novel predictions of previously unobserved phenomena: ungrokking and semi-grokking.

## 1 INTRODUCTION

Grokking is a puzzling but important phenomenon: even after a network achieves nearly perfect *training* loss but poor *test* loss, the test loss can dramatically improve with more training (Power et al., 2021). Understanding *why* this sudden jump in model capabilities happens would aid a scientific understanding of the generalisation behaviour of models produced by deep learning.

We present, formalise, and validate a theory of grokking. Despite much attention, previous attempts to explain grokking have not yet extended to a full theory. Some, for example, focus on relatively narrow special cases (Thilak et al., 2022) or find correlates of generalisation (Liu et al., 2022; 2023).

Our theory builds on a discovery by Nanda et al. (2023): grokking in modular arithmetic happens when the network switches between two emergent sub-networks, which we term "circuits" following Olah et al. (2020). The initial "memorising circuit" simply encodes examples seen so far and the later "generalising circuit" learns an approximation of the underlying maths. This raises the key question on which we focus: why does the generalising circuit develop and "take over" from the memorising one, which already achieves nearly perfect training loss?

The answer lies in the relative "efficiency" of different circuits. More efficient circuits achieve the same predictive loss with lower parameter norm. When multiple circuits achieve strong training performance, parameter regularisation (weight decay) selects for circuits with higher efficiency.

Memorising circuits can be very efficient for small datasets, while generalising circuits pay a larger up-front cost but scale better with dataset size. The generalising circuit lets the network encode new data points cheaply. In contrast, memorising circuits incur significant cost to encode each new datapoint. Even if the memorising circuit can achieve perfect accuracy, a regularised loss that values efficiency will prefer an accurate generalising solution for a large enough dataset. Common regularising losses like $L_2$ weight decay can be derived using minimum-description length principles as maximising the compression efficiency of a network given simple assumptions (Hinton & van Camp, 1993).

As a result, three ingredients in combination can cause grokking: (1) one circuit generalises well while another does not (2) the generalising circuit is more efficient (3) the generalising circuit is learned more slowly.

We validate this explanation empirically. We can quantify the efficiencies of generalising and memorising circuits as dataset size changes. We predict and validate the existence of a crossover critical dataset size, $D_{\text{crit}}$, at which the average efficiency of the generalising circuit is about the same

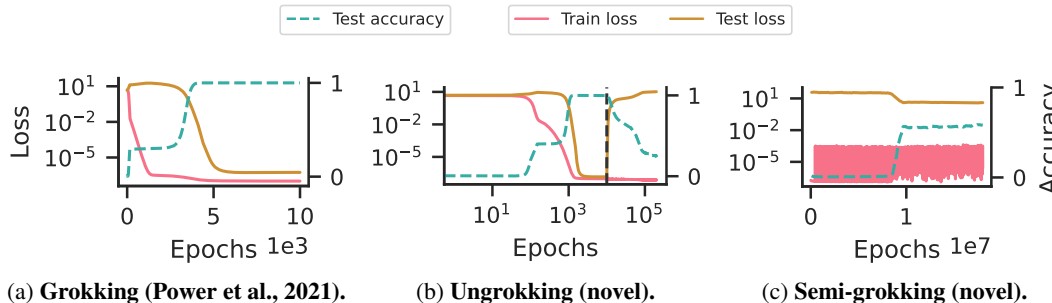

(a) **Grokking (Power et al., 2021).**     (b) **Ungrokking (novel).**     (c) **Semi-grokking (novel).**

Figure 1: (a) **Grokking**. Train loss (pink) falls quickly but test loss (gold) stays high. Later, test loss rapidly falls and 100% accuracy is achieved. (b) **Ungrokking (our novel phenomenon).** A fully grokked network is now trained on a subset of the original training set (at dotted line) below the critical dataset size. The network reverts to bad test loss/accuracy while train loss continues to improve. Note the log-scale x-axis. (c) **Semi-grokking (our novel phenomenon).** A randomly initialised network is trained on the critical dataset size. We see sharp delayed generalisation, but only to partial test accuracy ($\sim$60%).

as the memorising circuit. Using this, we predict and demonstrate two new behaviours (Figure 1). In *ungrokking*, a model that has successfully grokked returns to poor test accuracy when further trained on a subset of the original data much smaller than $D_{\text{crit}}$. In *semi-grokking*, we choose a dataset size near the critical dataset size where both circuits are similarly efficient, leading to a phase transition with only middling test accuracy.

We make the following contributions:

1. We explain how three ingredients can cause grokking (Section 2).
2. We explain why generalising circuits can be more efficient than memorising (Section 3.1).
3. Our theory implies a "critical dataset size" which we use to *predict* two novel phenomena: *semi-grokking* and *ungrokking* (Section 3.2).
4. We confirm our predictions empirically, providing support for the theory (Section 4).

## 2 THREE INGREDIENTS FOR GROKKING

The explanation of grokking which we elaborate and formalise builds on the presence of competing sub-networks, or circuits, which generalise to different extents (Nanda et al., 2023). In this section, we describe three ingredients which can jointly lead to grokking. In later sections, we discuss what might cause the ingredients to be present, and show how they appear in practice.

Suppose that we have just a memorising and a generalising circuit that are constituents of a neural network. In real networks, individual circuits are rarely entirely memorising or entirely generalising and there are many overlapping circuits. But here, for the sake of clear exposition, we consider cases like the one uncovered by Nanda et al. (2023) in which two distinct circuits are at opposite ends of that spectrum. Grokking phenomena can happen whenever the following ingredients are present:

1. **Generalising circuit:** There are two families of circuits that both achieve good training performance: a memorising circuit family with poor test performance, and a generalising circuit family with good test performance.
2. **Efficiency:** The generalising circuit is more "efficient" than the memorising circuit for marginal data, i.e., it can produce equal training predictive loss with a lower parameter norm.
3. **Slow vs fast learning:** The generalising circuit is learned more slowly than memorising, such that during early phases of training the memorising circuit contributes more to predictions than the generalising circuit.

Efficiency is an important conceptual tool for this explanation. We say that a circuit is more efficient if it can achieve the same predictive loss (e.g., cross-entropy) with a lower $L_2$ parameter norm (or

alternative regularising loss).[1] This can be interpreted as compression efficiency within the minimum description length framework given Gaussianity assumptions (Hinton & van Camp, 1993).

Predictive losses like cross-entropy encourage confident correct predictions—increasing the scale of the output logits allows a lower loss (see Theorem D.1). The weight-decay loss opposes this, penalising the larger weights that permit larger logits.

This balance of opposing forces means that optimisation selects for more efficient circuits. Consider that the most efficient circuit is the one that can produce a given predictive loss with the lowest parameter norm. When two circuits have the same confidently-large correct logits then the more efficient one will have lower overall loss—so gradient descent will tend to increase the contribution of that circuit to the network's overall prediction. In addition, during training there is a gradient pressure to produce accurate logits with a lower parameter norm by replacing a less efficient circuit with a more efficient one, which causes generalising circuits to emerge.

Overall, we can explain grokking as follows. Early in training, a memorising circuit quickly appears (ingredient 3) and is able to make good predictions on the training set. This leads to strong train performance and poor test performance. As training progresses, a sufficiently good generalising circuit emerges (ingredient 1). Because the generalising circuit is more efficient (ingredient 2), the model "reallocates" parameter-norm "budget" towards the generalising circuit, causing improved test performance.

In Appendix B.5 we describe the set-up for an illustrative example demonstrating these basic ingredients. The empirical results of this illustration are shown in Figure 2a, which should not be read as making any quantitative predictions about dynamics and is based on several simplifying design choices for sake of clarity and analytical tractability including not using neural networks. The illustration uses a weighted mixture model comprising a memorising and generalising circuit with different hypothetical parameter norms.

In Figure 2a we show how grokking can appear when all three ingredients are present. The memorising circuit is present initially and quickly achieves a high weight. The generalising circuit is learned more slowly but is more efficient (by hypothesis) and eventually its weight dominates leading to lower test loss. Figure 2b shows how the behaviour changes when it is not true that the generalising circuit is more efficient. In this case, there is never any pressure to switch from the memorising circuit and the test loss remains high. Last, Figure 2c shows how the behaviour changes when it is not true that the generalising circuit is learned more slowly. When the generalising circuit is present right at the beginning, the memorising circuit never needs to be used and no grokking is observed.

## 3    WHY GENERALISING CIRCUITS ARE MORE EFFICIENT

Section 2 demonstrated that grokking can arise when the generalising circuit is more efficient than the memorising circuit, but left open the question of *why*. In this section, we develop a theory based on training dataset size $D$, and use it to predict two new behaviours: *ungrokking* and *semi-grokking*.

### 3.1    RELATIONSHIP OF EFFICIENCY WITH DATASET SIZE

We should expect classifiers to generally get less efficient as dataset sizes grow. Consider a classifier $f_{\mathcal{D}}$ obtained by training optimally on a dataset $\mathcal{D}$ of size $D$ with weight decay, and a classifier $f_{\mathcal{D}'}$ obtained by training optimally on the same dataset with one additional point: $\mathcal{D}' = \mathcal{D} \cup \{(x, y^*)\}$. Intuitively, $f_{\mathcal{D}'}$ should not be *more* efficient than $f_{\mathcal{D}}$. Suppose it was more efficient—which means that for the same predictive loss it would have a lower parameter norm. But then the model $f_{\mathcal{D}'}$ would have been a lower-loss classifier for the original dataset than $f_{\mathcal{D}}$ was, a contradiction. So we should expect that, even when training is not always optimal, typically classifiers that learn larger datasets will be less efficient.

How does generalisation affect this picture? Let us suppose that $f_{\mathcal{D}}$ successfully generalises to predict $y^*$ for the new input $x$. Then, as we move from $\mathcal{D}$ to $\mathcal{D}'$, $\mathcal{L}_{\text{x-ent}}(f_{\mathcal{D}})$ likely does not worsen with this

---

[1]For brevity we focus on the common case where predictive loss is cross-entropy and it is regularised with $L_2$ weight decay. Our analysis extends to other predictive losses and regularisations (see Appendix D).

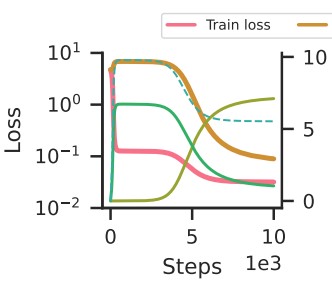 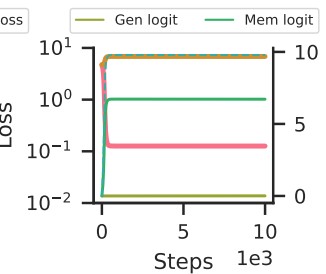 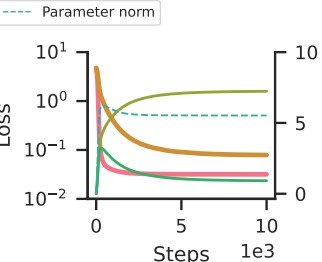

(a) **All three ingredients.** When the generalising circuit is more efficient but learned slower, we observe grokking. Total parameter norm falls due to the generalising circuit's higher efficiency.

(b) **Missing ingredient: Generalisation more efficient.** We set the hyperparameter controlling generalisation efficiency to be lower than memorisation efficiency. Since generalisation is also learned slower, it never grows, and test loss stays high due to memorisation throughout training.

(c) **Missing ingredient: Generalisation learned slower.** We set the hyperparameter controlling learning speed to be the same for generalisation and memorisation. The generalising circuit takes over from the very beginning due to its higher efficiency, and no grokking is observed.

Figure 2: **Generalisation must be learned slowly for grokking to arise.** We plot the results from our illustrative example (see Appendix B.5) showcasing the three ingredients for grokking: (1) a generalising and a memorising circuit, where (2) the generalising circuit is more efficient, but (3) slowly learned. We show that grokking does not occur when either of the last two ingredients are removed.

new data point. Thus, we could expect to see the same classifier arise, with the same average logit value, parameter norm, and efficiency.

Now suppose $f_{\mathcal{D}}$ instead *fails* to predict the new data point $(x, y^*)$. Then the classifier learned for $\mathcal{D}'$ will likely be *less* efficient: $\mathcal{L}_{\text{x-ent}}(f_{\mathcal{D}})$ would be much higher due to this new data point, and so the new classifier must incur some additional regularisation loss to reduce $\mathcal{L}_{\text{x-ent}}$ on the new point.

Applying this analysis to our circuits, we should expect the generalising circuit's efficiency to remain unchanged as $D$ increases arbitrarily high, since it does not need to change to accommodate new training examples. In contrast, the memorising circuit must change with nearly every new data point, and so we should expect its efficiency to decrease as $D$ increases. Thus, when $D$ is sufficiently large, we expect generalisation to be more efficient than memorisation. (Note however that when the set of possible inputs is small, even the maximal $D$ may not be "sufficiently large".)

**Critical threshold for dataset size.** Intuitively, we expect that for extremely small datasets (say, $D < 5$), it is extremely easy to memorise the training dataset. So, we hypothesise that for these very small datasets, the memorisation circuit is more efficient than the generalising one. However, as argued above, memorisation will get less efficient as $D$ increases, and so there will be a critical dataset size $D_{\text{crit}}$ at which memorisation and generalisation are approximately equally efficient. When $D \gg D_{\text{crit}}$, generalisation is more efficient and we expect grokking, and when $D \ll D_{\text{crit}}$, memorisation is more efficient and so grokking should not happen.

**Effect of weight decay on $D_{\text{crit}}$.** Since $D_{\text{crit}}$ is determined only by the relative efficiencies of the generalising and memorising circuits, and none of these depends on the exact value of weight decay (just on weight decay being present at all), our theory predicts that $D_{\text{crit}}$ should *not* change as a function of weight decay. Of course, the strength of weight decay may still affect other properties such as the number of epochs till grokking.

### 3.2 IMPLICATIONS OF CROSSOVER: UNGROKKING AND SEMI-GROKKING.

By thinking through the behaviour around the critical threshold for dataset size, we predict the existence of two phenomena that, to the best of our knowledge, have not previously been reported.

**Ungrokking.**  Suppose we take a network that has been trained on a dataset with $D > D_{\text{crit}}$ and has already exhibited grokking, and continue to train it on a smaller dataset with size $D' < D_{\text{crit}}$. In this new training setting, the memorising circuit is now more efficient than the generalising circuit, and so we predict that with enough further training gradient descent will reallocate weight from the former to the latter, leading to a transition from high test performance to low test performance. Since this is exactly the opposite observation as in regular grokking, we term this behaviour "ungrokking".

Ungrokking is superficially related to catastrophic forgetting (McCloskey & Cohen, 1989; Ratcliff, 1990), but with some large differences. Catastrophic forgetting involves training on a new dataset and forgetting the old, while in ungrokking the "new" dataset can just be a subset of the old dataset. Catastrophic forgetting also involves both a bad train and test loss on the old data, while ungrokking is just a bad test loss. The underlying mechanisms are different, for example since $D_{\text{crit}}$ does not depend on weight decay, we predict the amount of "forgetting" (i.e. the test accuracy at convergence) also does not depend on weight decay. Finally, unlike catastrophic forgetting, since ungrokking should only be expected once $D' < D_{\text{crit}}$, if we vary $D'$ we predict that there will be a sharp transition from very strong to near-random test accuracy (around $D_{\text{crit}}$).

**Semi-grokking.**  Suppose we train a network on a dataset with $D \approx D_{\text{crit}}$. Generalising and memorising circuits would be similarly efficient, and there are two possible cases for what we expect to observe (illustrated in Theorem D.4).

In the first case, gradient descent would select either a memorising or a generalising circuit, and then make it the maximal circuit. This could happen in a consistent manner (for example, perhaps since memorisation is learned faster it always becomes the maximal circuit), or in a manner dependent on the random initialisation. In either case we would simply observe the presence or absence of grokking.

In the second case, gradient descent would produce a mixture of both memorising and generalising circuits. Since neither memorisation nor generalisation would dominate the prediction on the test set, we would expect middling test performance.

Memorisation would still be learned faster, and so this would look similar to grokking: an initial phase with good train performance and bad test performance, followed by a transition to significantly improved test performance. Since we only get to middling generalisation unlike in typical grokking, we call this behaviour *semi-grokking*.

Our theory does not say which of the two cases will arise in practice, but in Section 4.3 we find that semi-grokking does happen in our setting.

## 4 EXPERIMENTAL EVIDENCE

Our explanation of grokking has some support from from prior work. **Generalising circuit:** Nanda et al. (2023, Figure 1) identify and characterise the generalising circuit learned at the end of grokking in the case of modular addition. **Slow vs fast learning:** Nanda et al. (2023, Figure 7) demonstrate "progress measures" showing that the generalising circuit develops and strengthens long after the network achieves perfect training accuracy in modular addition.

To further validate our explanation, we empirically test our predictions from Section 3:

(P1) **Efficiency:** We confirm our prediction that the generalising circuit efficiencies are independent of dataset size, while the memorising circuit efficiencies decrease as training dataset size increases.

(P2) **Ungrokking (phase transition):** We confirm our prediction that ungrokking shows a phase transition around $D_{\text{crit}}$.

(P3) **Ungrokking (weight decay):** We confirm our prediction that the final test accuracy after ungrokking is independent of the strength of weight decay.

(P4) **Semi-grokking:** We demonstrate that semi-grokking occurs in practice.

**Training details.** We train 1-layer Transformer models with the AdamW optimiser (Loshchilov & Hutter, 2019) on cross-entropy loss (see Appendix B for more details). All results in this section

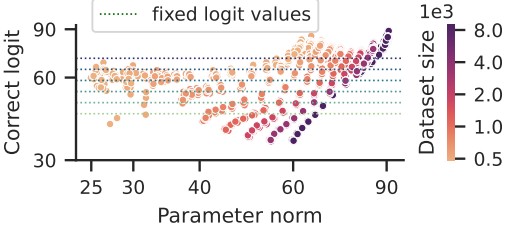

(a) **Memorisation scatter plot.** At a fixed logit value (dotted horizontal lines), parameter norm increases with dataset size.

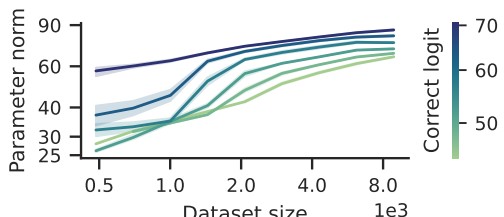

(b) **Memorisation isologit curves.** Curves go up and right, showing that parameter norm increases with dataset size when holding logits fixed.

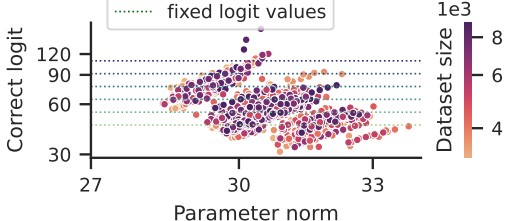

(c) **Generalisation scatter plot.** There is no obvious structure to the colours, suggesting that the logit to parameter norm relationship is independent of dataset size.

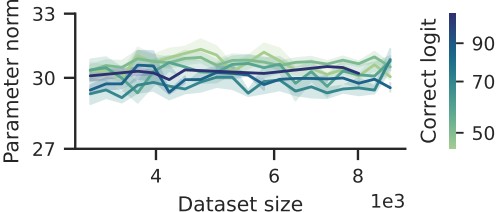

(d) **Generalisation isologit curves.** The curves are flat, showing that for fixed logit values the parameter norm does not depend on dataset size.

Figure 3: **Efficiency of memorisation-only and generalisation-only networks.** We collect and visualise a dataset of triples $(o^y, P_m, D)$ (correct logit, parameter norm, and dataset size), each corresponding to a training run with varying random seed, weight decay, and dataset size, for both memorising and generalising networks. Besides a standard scatter plot, we geometrically bucket logit values into six buckets, and plot "isologit curves" showing the dependence of parameter norm on dataset size for each bucket. The results validate our theory that (1) memorisation requires larger parameter norm to produce the same logits as dataset size increases, and (2) generalisation uses the same parameter norm to produce fixed logits, irrespective of dataset size. In addition, memorisation has a much wider range of parameter norms than generalisation, and at the extreme can be more efficient than generalisation.

are on the modular addition task ($a + b \mod P$ for $a, b \in (0, \ldots, P-1)$ and $P = 113$) unless otherwise stated; results on 9 additional tasks can be found in Appendix B.

## 4.1 RELATIONSHIP OF EFFICIENCY WITH DATASET SIZE

We first test our prediction about memorisation and generalisation efficiency:

**(P1) Efficiency.** We predict (Section 3.1) that memorisation efficiency decreases with increasing train dataset size, while generalisation efficiency stays constant.

To test (P1), we look at training runs where only one circuit is present, and see how the logits $o_i^y$ vary with the parameter norm $P_i$ (by varying the weight decay) and the dataset size $D$.

**Experiment setup.** We produce memorisation-only networks by using completely random labels for the training data (Zhang et al., 2021), and assume that the entire parameter norm at convergence is allocated to memorisation. We produce generalising-only networks by training on large dataset sizes and checking that $> 95\%$ of the logit norm comes from just the trigonometric subspace (see Appendix C for details).

**Results.** Figures 3a and 3b confirm our theoretical prediction for memorisation efficiency. Specifically, to produce a fixed logit value, a higher parameter norm is required when dataset size is increased, implying decreased efficiency. In addition, for a fixed dataset size, scaling up logits requires scaling up parameter norm, as expected. Figures 3c and 3d confirm our theoretical prediction for generalisation

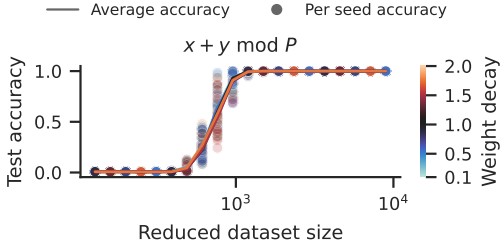

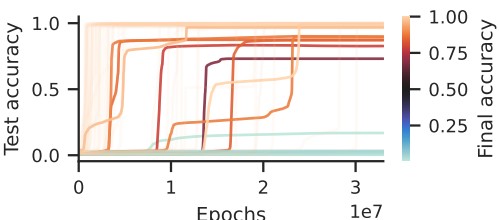

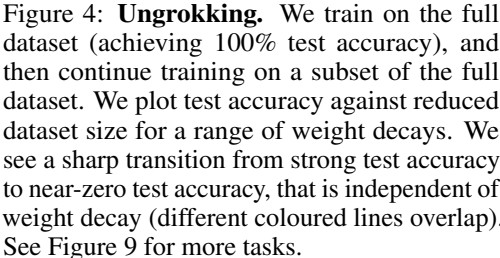

Figure 4: **Ungrokking.** We train on the full dataset (achieving 100% test accuracy), and then continue training on a subset of the full dataset. We plot test accuracy against reduced dataset size for a range of weight decays. We see a sharp transition from strong test accuracy to near-zero test accuracy, that is independent of weight decay (different coloured lines overlap). See Figure 9 for more tasks.

Figure 5: **Semi-grokking.** We plot test accuracy against training epochs for many training runs with varying dataset sizes. Of 200 runs, at least 6 show clear semi-grokking at the end of training. Many other runs show transient semi-grokking, hovering around middling test accuracy for millions of epochs, or having multiple plateaus, before fully generalising.

efficiency. To produce a fixed logit value, the same parameter norm is required irrespective of the dataset size.

Figure 3 shows significant variance across random seeds. We speculate that there are many different circuits implementing the same overall algorithm, but they have different efficiencies, and the random initialisation determines which one gradient descent finds. For example, in the case of modular addition, the generalising algorithm depends on a set of "key frequencies" (Nanda et al., 2023); different choices of key frequencies could lead to different efficiencies.

It may appear from Figure 3c that increasing parameter norm does not increase logit value, contradicting our theory. However, this is a statistical artefact caused by the variance from the random seed. We *do* see "stripes" of particular colours going up and right: these correspond to runs with the same seed and dataset size, but different weight decay, and they show that when the noise from the random seed is removed, increased parameter norm clearly leads to increased logits.

### 4.2 UNGROKKING: OVERFITTING AFTER GENERALISATION

We now turn to testing our predictions about ungrokking. Figure 1b demonstrates that ungrokking happens in practice. In this section we focus on testing that it has the properties we expect.

**(P2) Phase transition and (P3) weight decay.** We predict (Section 3.2) that if we plot test accuracy at convergence against the size of the reduced training dataset $D'$, there will be a phase transition around $D_{\text{crit}}$, and that test accuracy at convergence is independent of the strength of weight decay.

**Experiment setup.** We train a network to convergence on the full dataset to enable perfect generalisation, then continue training the model on a small subset of the full dataset, and measure the test accuracy at convergence. We vary both the size of the small subset, as well as the strength of the weight decay.

**Results.** Figure 4 shows the results, and clearly confirms both (P2) and (P3). Appendix B has additional results, and in particular Figure 9 replicates the results for many additional tasks.

### 4.3 SEMI-GROKKING: EVENLY MATCHED CIRCUITS

Unlike the previous predictions, semi-grokking is not strictly implied by our theory. However, as we will see, it turns out that it does occur in practice.

**(P4) Semi-grokking.** When training at around $D \approx D_{\text{crit}}$, where the memorisation and generalisation circuits have roughly equal efficiencies, the final network at convergence should either be entirely composed of the most efficient circuit, or of roughly equal proportions of memorisation and generalisation. If the latter, we should observe a transition to middling test accuracy well after near-perfect

train accuracy. We detail a number of difficulties with demonstrating semi-grokking in practice in Appendix B.1.

**Experiment setup.** We train 10 seeds for each of 20 dataset sizes evenly spaced in the range $[1500, 2050]$ (somewhat above our estimate of $D_{\text{crit}}$).

**Results.** Figure 1c shows an example of a single run that demonstrates semi-grokking, and Figure 5 shows test accuracies over time for every run. These validate our initial hypothesis that semi-grokking may be possible, but also raise new questions.

In Figure 1c, we see a phenomenon peculiar to semi-grokking: training loss fluctuates in a set range. We leave investigation of this to future work.

In Figure 5, we observe that there is often *transient* semi-grokking, where a run hovers around middling test accuracy for millions of epochs, or has multiple plateaus, before generalising perfectly. We speculate that each transition corresponds to gradient descent strengthening a new generalising circuit that is more efficient than any previously strengthened circuit, but took longer to learn. We would guess that if we had trained for longer, many of the semi-grokking runs would exhibit full grokking, and many of the runs that didn't generalise at all would generalise at least partially to show semi-grokking.

## 5 RELATED WORK

**Grokking.** There are many attempts to explain grokking (Power et al., 2021). We extend and build on initial explorations by Nanda et al. (2023, Appendix E) and Davies et al. (2023) which suggest that grokking is explained by a generalising circuit that is slowly learned but is favoured by inductive biases. We further operationalise the "inductive bias" argument by focusing on the relative *efficiency* at producing large logits with small parameter norm. In addition, we provide significant empirical support by predicting and verifying the existence of a critical threshold $D_{\text{crit}}$ and the novel phenomena ungrokking and semi-grokking.

Other explanations have focused on narrow situations, such as cases where oscillations (Notsawo Jr et al., 2023) or "slingshots" (Thilak et al., 2022) are present (which are not required or present in any of our experiments). Alternatively, other works provide partial explanations which are consistent with our theory but do not explain all the phenomena. For example, Liu et al. (2022) relate perfect generalisation to sufficient data for representation learning, which does not predict or explain semi-grokking. Liu et al. (2023) relate early memorisation to large parameter initialisation, which is consistent with our theory but does not explain or predict the dynamics that our theory discovers.

The metrics for circuit strength which we use depend on both Nanda et al. (2023) and Chughtai et al. (2023). Merrill et al. (2023) show similar results on sparse parity: in particular, they show that a sparse subnetwork is responsible for the well-generalising logits, and that it grows as grokking happens.

**Weight decay.** It is widely known that weight decay can improve generalisation (Krogh & Hertz, 1991), though the mechanisms for this effect are poorly understood (Zhang et al., 2018). One hypothesis is that weight decay is equivalent to using a minimum description length principle, assuming Gaussian distributions (Hinton & van Camp, 1993). This supports our model of generalising circuits as being preferred by weight decay because they are more efficient than memorising circuits at large dataset sizes.

**Understanding deep learning through circuit-based analysis.** One goal of *interpretability* is to understand the internal mechanisms by which neural networks exhibit specific behaviours (Olah et al., 2020; Elhage et al., 2021; Erhan et al., 2009; Meng et al., 2022; Cammarata et al., 2021; Wang et al., 2022; Li et al., 2022; Geva et al., 2020). Such work can also be used to understand deep learning.

Olsson et al. (2022) explain a phase change in the training of language models by reference to *induction heads*, a family of circuits that produce in-context learning. In concurrent work, Singh et al. show that the in-context learning from induction heads is later replaced by in-weights learning in the absence of weight decay, but remains strong when weight decay is present. We hypothesise that this effect is also explained through circuit efficiency: the in-context learning from induction heads is a generalising algorithm and so is favoured by weight decay given a large enough dataset size.

## 6 DISCUSSION

Most of our analysis focuses on weight decay regularisation with $L_2$ norms (although in Appendix D we prove results for a wider class of losses). Grokking has been observed even when weight decay is not present (Power et al., 2021; Thilak et al., 2022) though it is slower and often much harder to elicit (Nanda et al., 2023, Appendix D.1). This shows that our explanation is incomplete and at least some other forces are able to cause grokking.

However, we do not think this implies that the explanation is incorrect. We hypothesise that other regularisers similarly favour generalisation circuits over memorisation circuits, such as the implicit regularisation of gradient descent (Soudry et al., 2018; Lyu & Li, 2019; Wang et al., 2021; Smith & Le, 2017), and that the speed of the transition from memorisation to generalisation is based on the *sum* of these effects and the effect from weight decay. Although our explanations focus on weight decay, the important part is that there is a systematic preference for the generalising circuit during training, which might be provided by other sources of regularisation. This would explain why grokking takes longer as weight decay decreases (Power et al., 2021), and does not completely vanish in the absence of weight decay. Given that there is a potential extension of our theory that explains grokking without weight decay, and the significant confirming evidence that we have found for our theory in settings with weight decay, we are overall confident that our explanation is at least one part of the true explanation when weight decay is present.

In addition, there are many constraints other than parameter norm that might affect the circuit selection: fitting the training data, capacity in "bottleneck activations" (Elhage et al., 2021), interference between circuits (Elhage et al., 2022), and more. This may limit the broader applicability of our theory, despite its success in explaining grokking.

**Broader applicability: realistic settings.** We expect that the general concepts of circuits, efficiency, and speed of learning continue to apply. However, in realistic settings, good performance would result from different circuit families that contribute different aspects (e.g. language modelling requires spelling, grammar, arithmetic, etc). We expect that these will have a wide continuum of learning speeds and efficiencies. In contrast, for grokking in "algorithmic" tasks like modular arithmetic, we explain the sharp transition occurring due to a shift from a memorising to a generalising cluster, with no intermediate circuits in between. We demonstrate this difference with experiments on MNIST (LeCun et al., 1998) in Figures 7 and 10.

**Future work.** Within grokking, several interesting puzzles are still left unexplained. Why does the time taken to grok rise super-exponentially as dataset size decreases? How does the random initialisation interact with efficiency to determine which circuits are found by gradient descent? What causes generalising circuits to develop slower? Investigating these puzzles is a promising avenue for further work.

While the direct application of our work is to understand the puzzle of grokking, we are excited about the potential for understanding deep learning more broadly through the lens of circuit efficiency. We would be excited to see work looking at the role of circuit efficiency in more realistic settings, and work that extends circuit efficiency to consider other constraints that gradient descent must navigate.

## 7 CONCLUSION

The central question of our paper is: in grokking, why does the network's test performance improve dramatically upon continued training, having already achieved nearly perfect training performance? Our explanation is: the generalising solution is more "efficient" but slower to learn than the memorising solution. After quickly learning the memorising circuit, gradient descent can still decrease loss even further by simultaneously strengthening the efficient, generalising circuit and weakening the inefficient, memorising circuit.

Based on our theory we predict and demonstrate two novel behaviours: *ungrokking*, in which a model that has perfect generalisation returns to memorisation when it is further trained on a dataset with size smaller than the critical threshold, and *semi-grokking*, where we train a randomly initialised network on the critical dataset size which results in a grokking-like transition to middling test accuracy. Our explanation is the only one we are aware of that has made (and confirmed) such surprising advance predictions, and we have significant confidence in the explanation as a result.

## REPRODUCIBILITY STATEMENT

We provide information about the datasets, setup, and hyperparameters required to reproduce our experiments and minimal example in Appendix B. We describe difficulties with eliciting semi-grokking in particular, and how we overcome them. We state and explain all proofs along with their assumptions in Appendix D.

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

# A  NOTATION

We consider classification using deep neural networks under the cross-entropy loss. In particular, we are given a set of inputs $X$, a set of labels $Y$, and a training dataset, $\mathcal{D} = \{(x_1, y_1^*), \ldots (x_D, y_D^*)\}$.

For an arbitrary classifier $f : X \times Y \to \mathbb{R}$, the softmax cross entropy loss is given by:

$$\mathcal{L}_{\text{x-ent}}(f) = -\frac{1}{D} \sum_{(x,y^*) \in \mathcal{D}} \log \frac{\exp(f(x, y^*))}{\sum_{y' \in Y} \exp(f(x, y'))}. \tag{1}$$

The output of a classifier for a specific class is the class *logit*, denoted by $o_f^y(x) := f(x, y)$. When the input $x$ is clear from context, we will denote the logit as $o_f^y$. We denote the vector of the logits for all classes for a given input as $\vec{o}_f(x)$ or $\vec{o}_f$ when $x$ is clear from context.

Parametric classifiers (such as neural networks) are parameterised with a vector $\theta$ that induces a classifier $f_\theta$. The parameter norm of the classifier is $P_{f_\theta} := \|\theta\|$. It is common to add *weight decay* regularisation, which is an additional loss term $\mathcal{L}_{\text{wd}}(f_\theta) = \frac{1}{2}(P_{f_\theta})^2$. The overall loss is given by

$$\mathcal{L}(f_\theta) = \mathcal{L}_{\text{x-ent}}(f) + \alpha \mathcal{L}_{\text{wd}}(f_\theta), \tag{2}$$

where $\alpha$ is a constant that trades off between softmax cross entropy and weight decay.

**Circuits.**  Inspired by Olah et al. (2020), we use the term *circuit* to refer to an internal mechanism by which a neural network works. We only consider circuits that map inputs to logits, so that a circuit $C$ induces a classifier $f_C$ for the overall task. We elide this distinction and simply write $C$ to refer to $f_C$, so that the logits are $o_C^y$, the loss is $\mathcal{L}(C)$, and the parameter norm is $P_C$.

For any given algorithm, there exist multiple circuits that implement that algorithm. Abusing notation, we use generalising (memorising) circuit to refer either to the *family* of circuits that implements the generalising (memorising) algorithm, or a single circuit from the appropriate family.

# B  EXPERIMENTAL DETAILS AND MORE EVIDENCE

For all our experiments, we use 1-layer decoder-only transformer networks (Vaswani et al., 2017) with learned positional embeddings, untied embeddings/unembeddings, The hyperparameters are as follows: $d_{\text{model}} = 128$ is the residual stream width, $d_{\text{head}} = 32$ is the size of the query, key, and value vectors for each attention head, $d_{\text{mlp}} = 512$ is the number of neurons in the hidden layer of the MLP, and we have $d_{\text{model}}/d_{\text{head}} = 4$ heads per self-attention layer. We optimise the network with full batch training (that is, using the entire training dataset for each update) using the AdamW optimiser (Loshchilov & Hutter, 2019) with $\beta_1 = 0.9$, $\beta_2 = 0.98$, learning rate of $10^{-3}$, and weight decay of $1.0$. In some of our experiments we vary the weight decay in order to produce networks with varying parameter norm.

Following Power et al. (2021), for a binary operation $x \circ y$, we construct a dataset of the form $\langle x \rangle \langle \circ \rangle \langle y \rangle \langle = \rangle \langle x \circ y \rangle$, where $\langle a \rangle$ stands for the token corresponding to the element $a$. We choose a fraction of this dataset at random as the train dataset, and the remainder as the test dataset. The first 4 tokens $\langle x \rangle \langle \circ \rangle \langle y \rangle \langle = \rangle$ are the input to the network, and we train with cross-entropy loss over the final token $\langle x \circ y \rangle$. For all modular arithmetic tasks we use the modulus $p = 113$, so for example the size of the full dataset for modular addition is $p^2 = 12769$, and $d_{\text{vocab}} = 115$, including the $\langle + \rangle$ and $\langle = \rangle$ tokens.

## B.1  SEMI-GROKKING

There are a number of difficulties in demonstrating an example of semi-grokking in practice. First, the time to grok increases super-exponentially as the dataset size $D$ decreases (Power et al., 2021, Figure 1), and $D_{\text{crit}}$ is significantly smaller than the smallest dataset size at which grokking has been demonstrated. Second, the random seed causes significant variance in the efficiency of the generalising and memorising circuits, which in turn affects $D_{\text{crit}}$ for that run. Third, accuracy changes sharply with the ratio of strengths of generalisation to memorisation (Figure 12). To observe a transition to middling accuracy, we need to have balanced generalising and memorising circuit

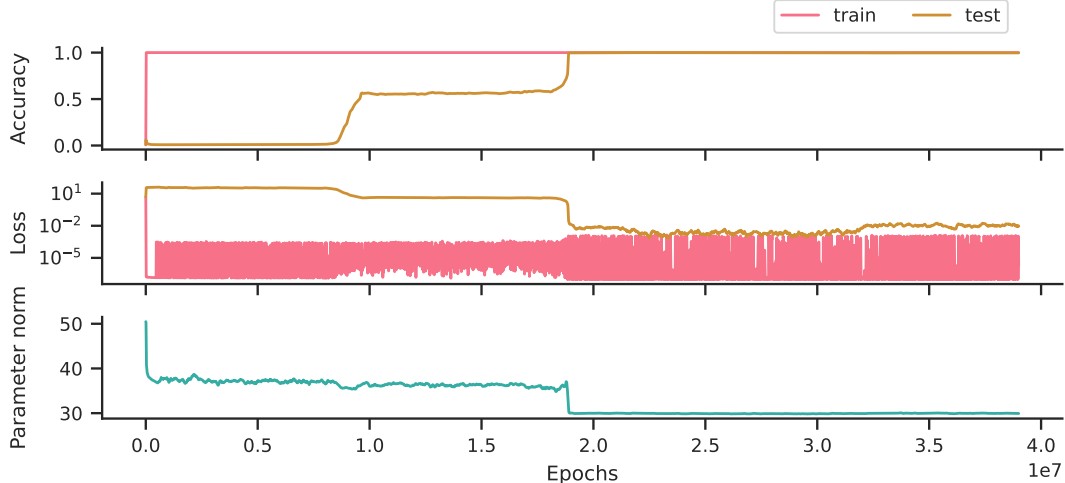

Figure 6: **Examining a single semi-grokking run in detail.** We plot accuracy, loss, and parameter norm over training for a single cherry-picked modular addition run at a dataset size of 1532 (12% of the full dataset). This run shows transient semi-grokking. At epoch $0.8 \times 10^7$, test accuracy rises to around 0.55, and then stays there for $10^7$ epochs, because generalising and memorising circuit efficiencies are balanced. At epoch $1.8 \times 10^7$, we speculate that gradient descent finds an even more efficient generalising circuit, as parameter norm drops suddenly and test accuracy rises to 1. At epoch $3.2 \times 10^7$ we see test loss *rise*, we do not know why. There seem to be multiple phases, perhaps corresponding to the network transitioning between mixtures of multiple circuits with increasing efficiencies, but further investigation is needed.

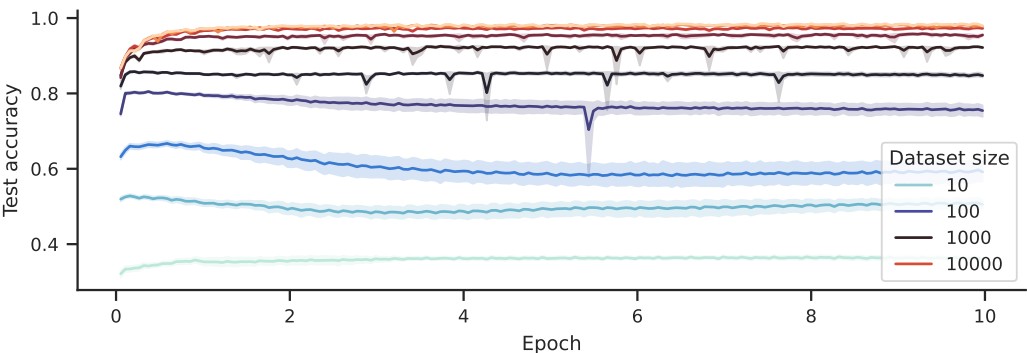

Figure 7: **Attempting semi-grokking on MNIST.** We train a small CNN on MNIST at various dataset sizes. Unlike in modular arithmetic, we see immediate rather than delayed generalisation to partial test accuracy. Increasing the dataset size smoothly and consistently increases the test accuracy (rather than showing a phase change). This is explained by a spectrum of feature-circuits, which form at increasing dataset sizes, and which cumulatively lead to smoothly increasing generalisation.

outputs, but this is difficult to arrange due to the variance with random seed. To address these challenges, we run many different training runs, on dataset sizes slightly *above* our best estimate of the typical $D_{\text{crit}}$, such that some of the runs will (through random noise) have an unusually inefficient generalising circuit or an unusually efficient memorising circuit, such that the efficiencies match and there is a chance to semi-grok.

Given the difficulty of demonstrating semi-grokking, we only run this experiment on modular addition. However, our experience with modular addition shows that if we only care about values at convergence, we can find them much faster by ungrokking from a grokked network (instead of semi-grokking from a randomly initialised network). Thus the ungrokking results on other tasks (Figure 9) provide some support that we would see semi-grokking on those tasks as well.

In Section 4.3 we looked at all semi-grokking training runs in Figure 5. Here, we investigate a single example of transient semi-grokking in more detail (see Figure 6). We speculate that there are multiple circuits with increasing efficiencies that generalise, and in these cases the more efficient circuits are slower to learn. This would explain transient semi-grokking: gradient descent first finds a less efficient generalising circuit and we see partial test accuracy, but since we are using the upper range of $D_{\text{crit}}$, eventually gradient descent finds a more efficient generalising circuit leading to full test accuracy.

## B.2  UNGROKKING

In Figure 8, we show many ungrokking runs for modular addition, and in Figure 9 we show ungrokking across many other tasks.

We have already seen that $D_{\text{crit}}$ is affected by the random initialisation. It is interesting to compare $D_{\text{crit}}$ when starting with a given random initialisation, and when ungrokking from a network that was trained to full generalisation with the same random initialisation. Figure 5 shows a semi-grokking run that achieves a test accuracy of $\sim 0.7$ with a dataset size of $\sim 2000$, while Figure 8 shows ungrokking runs that achieve a test accuracy of $\sim 0.7$ with a dataset size of around 800–1000, less than half of what the semi-grokking run required.

In Figure 12b, the final test accuracy after *ungrokking* shows a smooth relationship with dataset size, which we might expect if the generalising circuit is getting stronger on a smoothly increasing number of inputs compared to the memorising circuit. However due to the difficulties discussed previously, we don't see a smooth relationship between test accuracy and dataset size in *semi-grokking*.

These results suggest that $D_{\text{crit}}$ is an oversimplified concept, because in reality the initialisation and training dynamics affect which circuits are found, and therefore the dataset size at which we see middling generalisation.

## B.3  GENERALISING AND MEMORISING CIRCUIT DEVELOPMENT DURING GROKKING

In Figure 11 we show generalising and memorising circuit development via the proxy measures defined in Appendix C for a randomly-picked grokking run. Looking at these measures was very useful to form a working theory for why grokking happens. However as we note in Appendix C, these proxy measures tend to overestimate generalisation and underestimate memorisation.

We note some interesting phenomena in Figure 11:

1. Between epochs 200 to 1500, *both* the generalisation and memorisation logits are rising while parameter norm is falling, indicating that gradient descent is improving efficiency (possibly by removing irrelevant parameters).

2. After epoch 4000, the generalisation logit *falls* while the memorisation logit is already $\sim 0$. Since test loss continues to fall, we expect that incorrect logits from the memorising circuit on the test dataset are getting cleaned up, as described in Nanda et al. (2023).

## B.4  TRADEOFFS BETWEEN GENERALISATION AND MEMORISATION

In Section 3.1 we looked at the efficiency of generalisation-only and memorisation-only circuits. In this section we train on varying dataset sizes so that the network develops a mixture of generalising

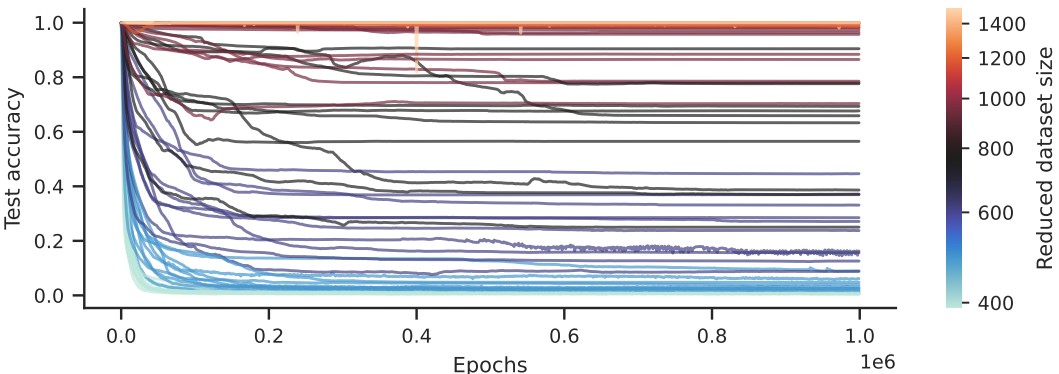

Figure 8: **Many ungrokking runs.** We show test accuracy over epochs for a range of ungrokking runs for modular addition. Each line represents a single run, and we sweep over 7 geometrically spaced dataset sizes in $[390, 1494]$ with 10 seeds each. Each run is initialised with parameters from a network trained on the full dataset (the initialisation runs are not shown), so test accuracy starts at 1 for all runs. When the dataset size is small enough, the network ungroks to poor test accuracy, while train accuracy remains at 1 (not shown). For an intermediate dataset size, we see ungrokking to middling test accuracy as generalising and memorising circuit efficiencies are balanced.

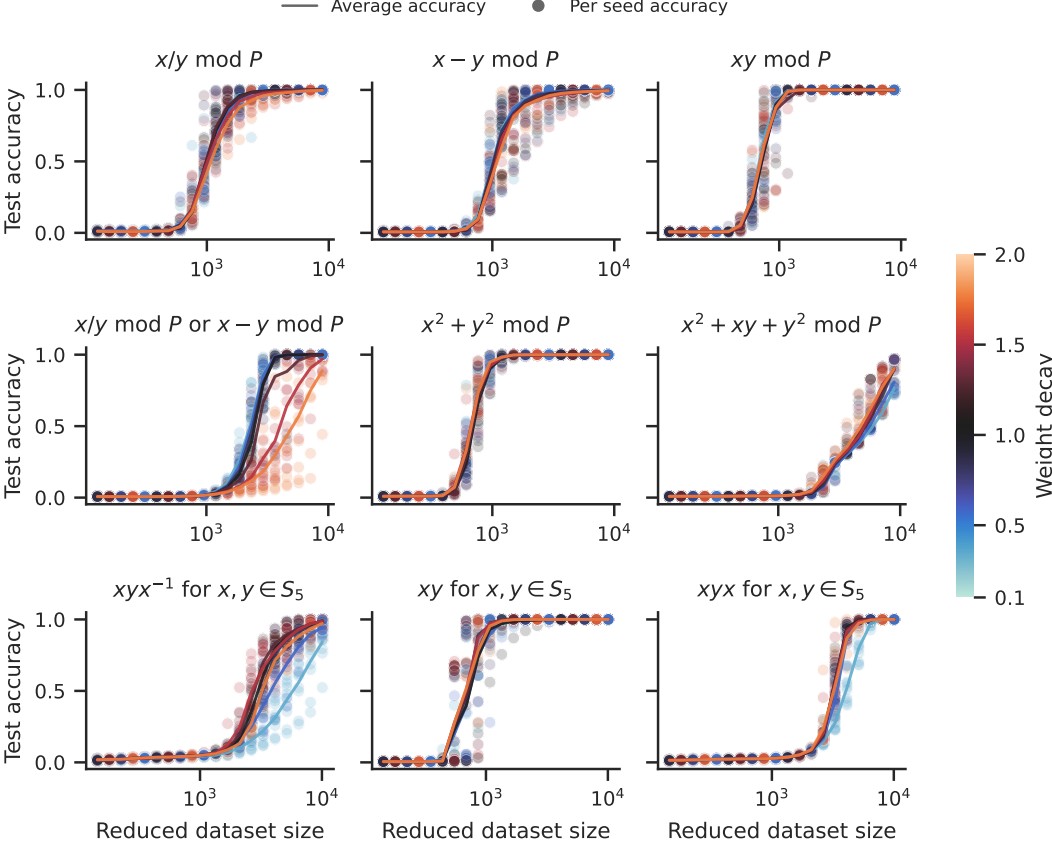

Figure 9: **Ungrokking on many other tasks.** We plot test accuracy against reduced dataset size for many other modular arithmetic and symmetric group tasks (Power et al., 2021). For each run, we train on the full dataset (achieving 100% accuracy), and then further train on a reduced subset of the dataset for 100k steps. The results show clear ungrokking, since in many cases test accuracy falls below 100%, often to nearly 0%. For most datasets the transition point is independent of weight decay (different coloured lines almost perfectly overlap).

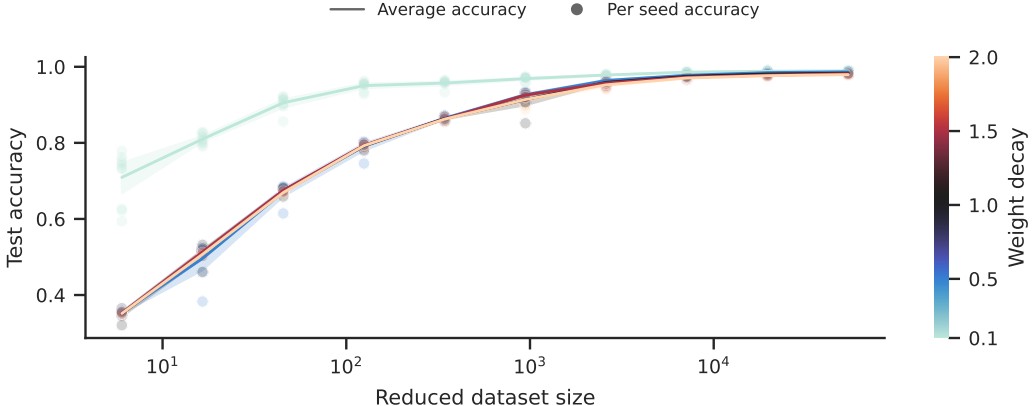

Figure 10: **Attempting ungrokking on MNIST.** When we try to exhibit ungrokking on MNIST, we instead see a smooth decline in test accuracy as dataset size reduces, consistent with a continuous spectrum of generalising-feature circuits, each of which forms at a fixed dataset size. Contrast this with Figure 9 where there is usually a critical dataset size at which we see a sharp change. Since each circuit's efficiency relative to memorisation depends only on the dataset size and not the weight decay, the aggregate behaviour is also independent of weight decay (different coloured lines almost perfectly overlap). The exception is at a weight decay of $0.1$ because retraining does not converge in 10 epochs here.

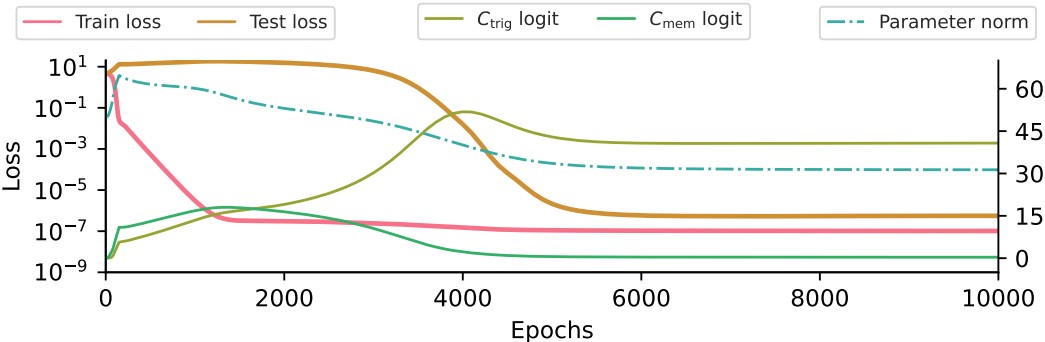

Figure 11: **Grokking occurs because the generalising circuit is more efficient than the memorising one.** We show loss, parameter norm, and the value of the correct logit for generalisation and memorisation for a randomly-picked training run. By step 200, the train accuracy is already perfect (not shown), train loss is low while test loss has risen, and parameter norm is at its maximum value, indicating strong memorisation. Train loss continues to fall rapidly until step 1500, as parameter norm falls and the generalising logit becomes higher than the memorising logit. At step 3500, test loss starts to fall as the high generalising logit starts to dominate, and by step 6000 we get good generalisation.

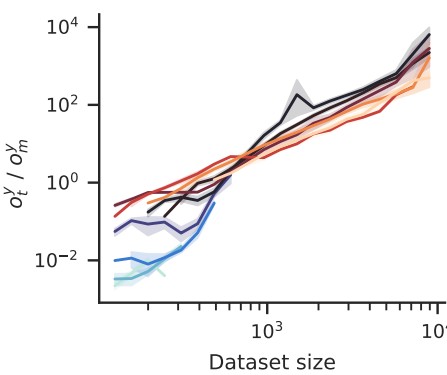 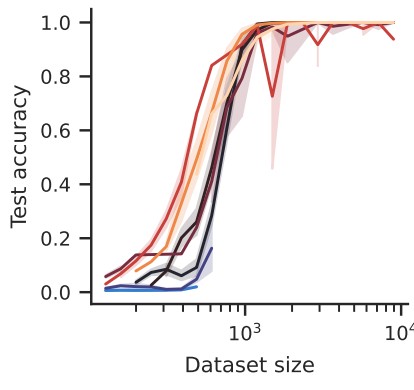 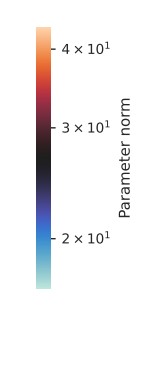

(a) **Logit ratio ($o_t^y / o_m^y$) vs dataset size ($D$).** Colours correspond to different bucketed values of parameter norm ($P$). Each line shows that as dataset size increases, a fixed parameter norm (fixed colour) is being reallocated smoothly towards increasing the trigonometric logit compared to the memorisation logit.

(b) **Test accuracy vs dataset size ($D$).** We see a smooth dependence on dataset size. Each line shows that as dataset size increases, the reallocation of a fixed parameter norm (fixed colour) towards the generalisation circuit from memorisation results in increasing accuracy.

Figure 12: **Relative strength at convergence.** We report logit ratios and test accuracy at convergence across a range of training runs, generated by sweeping over the weight decay and random seed to obtain different parameter norms at the same dataset size. We use the ungrokking runs from Figure 8, so every run is initialised with parameters obtained by training on the full dataset.

and memorising circuits, and study their relative strength using the correct logit as a proxy measure (described in Appendix C).

As we demonstrated previously, memorisation's efficiency drops with increasing dataset size, while generalisation's stays constant. Theorem D.4 (case 2) suggests that parameter norm allocated to a circuit is proportional to efficiency, and since logit values also increase with parameter norm, this implies that the ratio of the generalisation to memorisation logit $o_t^y / o_m^y$ should increase monotonically with dataset size.

In Figure 12a we see exactly this: the logit ratio changes monotonically (over 6 orders of magnitude) with increasing dataset size.

Due to the difficulties in training to convergence at small dataset sizes, we initialised all parameters from a generalisation-only network trained on the full dataset. We confirmed that in all the runs, at convergence, the training loss was lower than the training loss from a randomly initialised network, indicating that this initialisation allows our optimiser to find a better minimum than from random initialisation.

### B.5 ILLUSTRATIVE EXAMPLE

We describe a minimal example showing how the the three ingredients (described in Section 2) can lead to grokking. This should be read as an illustration of the ingredients, rather than making quantitative predictions about dynamics. We make several simplifying design choices to make the example clear and tractable. For analytical tractability, we let the generalisation and memorisation circuits be weighted input-output lookup tables, rather than internal circuits in neural networks.

**Set up.** For our illustration, we want to contrast an idealised generalising circuit with an idealised memorising circuit. An idealised memorising circuit makes perfect predictions for all points in the training data but incorrect predictions elsewhere. Meanwhile an idealised generalising circuit makes perfect predictions for all points in both training and test data. We can formally express this by writing the output logits $o_f^y(x)$—defined with respect to the input, $x$, label, $y$, and classifier, $f$—for

each of the two circuits as indicator functions of the datasets that they have perfect performance on.

$$o_G^y(x) = \mathbb{1}\left[(x,y) \in \mathcal{D} \text{ or } (x,y) \in \mathcal{D}_{\text{test}}\right] \tag{3}$$

$$o_M^y(x) = \mathbb{1}\left[(x,y) \in \mathcal{D} \text{ or } (x,y) \in \mathcal{D}_{\text{mem}}\right]. \tag{4}$$

Here, the training dataset is $\mathcal{D}$; the classifier is evaluated on the test dataset, $\mathcal{D}_{\text{test}}$; and $\mathcal{D}_{\text{mem}}$ is a set of datapoints with the same $x$ as the test dataset but with confident wrong answers. In a real network, a generalising circuit and memorising circuit would not have such extreme performance.

In a neural network, multiple circuits combine to produce the final output. In a similar way, and allowing us to model learning, we look at the weighted combination of these two circuits.

$$o^y(x) = w_G o_G^y(x) + w_M o_M^y(x)$$

Unfortunately, if we learn $w_G$ and $w_M$ directly with gradient descent, we have no control over the *speed* at which the weights are learned. Inspired by Jermyn & Shlegeris (2022), we instead compute weights as multiples of two "subweights", and then learn the subweights with gradient descent. More precisely, we let $w_G = w_{G_1} w_{G_2}$ and $w_M = w_{M_1} w_{M_2}$, and update each subweight according to $w_i \leftarrow w_i - \lambda \cdot \partial \mathcal{L}/\partial w_i$. The speed at which the weights are strengthened by gradient descent can then be controlled by the initial values of the weights.

We can also model the "parameter norm" and its relationship to the relative weights of the two circuits. Because the circuits here are non-parametric idealisations we need to give an illustrative hyperparameter for each describing a hypothetical parameter norm, $P_G$ and $P_M$, which we define as the parameter norms when $w_M = w_G = 1$. By hypothesis, because the generalising circuit is more efficient, $P_G < P_M$ (ingredient 2).

When one of the circuits is better for prediction, its weight in the mixture will increase. This corresponds intuitively to the weights of a neural network scaling up so that the final logits scale up. In a $\kappa$-layer MLP with Relu activations and without biases, scaling all parameters by a constant $c$ would scale the outputs by $c^\kappa$. Inspired by this, we model the parameter norm of the generalising circuit as $w_G^{1/\kappa} P_G$ for some $\kappa > 0$, and similarly for the memorising circuit.

**Theoretical analysis.** We first analyse the optimal solutions to the setup above. We can ignore the subweights, as they only affect the speed of learning: $\mathcal{L}_{\text{x-ent}}$ and $\mathcal{L}_{\text{wd}}$ depend only on the weights, not subweights. Intuitively, to get minimal loss, we must assign higher weights to more efficient circuits – but it is unclear whether we should assign *no* weight to less efficient circuits, or merely smaller but still non-zero weights. Theorem D.4 shows that in our example, both of these cases can arise: which one we get depends on the value of $\kappa$.

In Figure 2 we show that two ingredients: multiple circuits with different efficiencies, and slow and fast circuit development, are sufficient to reproduce learning curves that qualitatively demonstrate grokking. In Table 1 we provide details about the simulation used to produce this figure.

## C    GENERALISATION AND MEMORISATION IN MODULAR ADDITION

In modular addition, given two integers $a, b$ and a modulus $p$ as input, where $0 \le a, b < p$, the task is to predict $a + b \mod p$. Nanda et al. (2023) identified the generalising algorithm implemented by a 1-layer transformer after grokking (visualised in Figure 13), which we call the "trigonometric" algorithm. In this section we summarise the algorithm, and explain how we produce our proxy metrics for the strength of the generalising and memorising circuits.

**Trigonometric logits.** We explain the structure of the logits produced by the trigonometric algorithm. For each possible label $c \in \{0, 1, \ldots p - 1\}$, the trigonometric logit $o^c$ will be given by $\sum_{\omega_k} \cos(\omega_k(a + b - c))$, for a few key frequencies $\omega_k = 2\pi \frac{k}{p}$ with integer $k$. For the true label $c^* = a + b \mod p$, the term $\omega_k(a + b - c^*)$ is an integer multiple of $2\pi$, and so $\cos(\omega_k(a + b - c^*)) = 1$. For any incorrect label $c \ne a + b \mod p$, it is very likely that at least *some* of the key frequencies satisfy $\cos(\omega_k(a + b - c)) \ll 1$, creating a large difference between $o^c$ and $o^{c^*}$.

**Trigonometric algorithm.** There is a set of key frequencies $\omega_k$. (These frequencies are typically whichever frequencies were highest at the time of random initialisation.) For an arbitrary label $c$, the logit $o^c$ is computed as follows:

Table 1: Hyperparameters used for our simulations.

(a) Generalisation is learned slower but is more efficient than memorisation.

| Parameter | Value |
|---|---|
| $P_g$ | 1 |
| $P_m$ | 2 |
| $\kappa$ | 1.2 |
| $\alpha$ | 0.005 |
| $w_{g_1}(0)$ | 0 |
| $w_{g_2}(0)$ | 0.005 |
| $w_{m_1}(0)$ | 0 |
| $w_{m_2}(0)$ | 1 |
| $q$ | 113 |
| $\lambda$ | 0.01 |

(b) Generalisation less efficient than memorisation.

| Parameter | Value |
|---|---|
| $P_g$ | 4 |
| $P_m$ | 2 |
| $\kappa$ | 1.2 |
| $\alpha$ | 0.005 |
| $w_{g_1}(0)$ | 0 |
| $w_{g_2}(0)$ | 0.005 |
| $w_{m_1}(0)$ | 0 |
| $w_{m_2}(0)$ | 1 |
| $q$ | 113 |
| $\lambda$ | 0.01 |

(c) Generalisation and memorisation learned at equal speeds.

| Parameter | Value |
|---|---|
| $P_g$ | 1 |
| $P_m$ | 2 |
| $\kappa$ | 1.2 |
| $\alpha$ | 0.005 |
| $w_{g_1}(0)$ | 0 |
| $w_{g_2}(0)$ | 1 |
| $w_{m_1}(0)$ | 0 |
| $w_{m_2}(0)$ | 1 |
| $q$ | 113 |
| $\lambda$ | 0.01 |

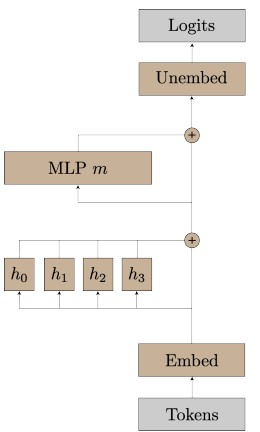

Computes logits using further trig identities:
$$\text{Logit}(c) \propto \cos(w(a+b-c))$$
$$= \cos(w(a+b))\cos(wc) + \sin(w(a+b))\sin(wc)$$

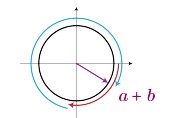

Calculates sine and cosine of $a+b$ using trig identities:
$$\sin(w(a+b)) = \sin(wa)\cos(wb) + \cos(wa)\sin(wb)$$
$$\cos(w(a+b)) = \cos(wa)\cos(wb) - \sin(wa)\sin(wb)$$

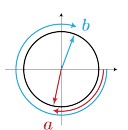

Translates one-hot a, b to Fourier basis:
$$a \to \sin(wa), \cos(wa)$$
$$b \to \sin(wb), \cos(wb)$$

Figure 13: **The trigonometric algorithm for modular arithmetic** (reproduced from Nanda et al. (2023)). Given two numbers $a$ and $b$, the model projects each point onto a corresponding rotation using its embedding matrix. Using its attention and MLP layers, it then composes the rotations to get a representation of $a + b \mod p$. Finally, it "reads off" the logits for each $c \in \{0, 1, \ldots, p-1\}$, by rotating by $-c$ to get $cos(\omega(a+b-c))$, which is maximised when $a + b \equiv c \mod P$ (since $\omega$ is a multiple of $2\pi$).

1. Embed the one-hot encoded number $a$ to $\sin(\omega_k a)$ and $\cos(\omega_k a)$ for the various frequencies $\omega_k$. Do the same for $b$.

2. Compute $\cos(\omega_k(a+b))$ and $\sin(\omega_k(a+b))$ using the intermediate attention and MLP layers via the trigonometric identities:

$$\cos(\omega_k(a+b)) = \cos(\omega_k a)\cos(\omega_k a) - \sin(\omega_k a)\sin(\omega_k b)$$
$$\sin(\omega_k(a+b)) = \sin(\omega_k a)\cos(\omega_k b) + \cos(\omega_k a)\sin(\omega_k b)$$

3. Use the output and unembedding matrices to implement the trigonometric identity:

$$o^c = \sum_{\omega_k}\cos(\omega_k(a+b-c)) = \sum_{\omega_k}\cos(\omega_k(a+b))\cos(\omega_k c) + \sin(\omega_k(a+b))\sin(\omega_k c).$$

**Isolating trigonometric logits.** Given a classifier $f$, we can aggregate its logits on every possible input, resulting in a vector $\vec{Z}_f$ of length $p^3$ where $\vec{Z}_f^{a,b,c} = o_f^c(\text{"}a + b =\text{"})$ is the logit for label $c$ on the input $(a, b)$. We are interested in identifying the contribution of the trigonometric algorithm to $\vec{Z}_f$.

We use the same method as Chughtai et al. (2023) and restrict $\vec{Z}_f$ to a much smaller trigonometric subspace.

For a frequency $\omega_k$, let us define the $p^3$-dimensional vector $\vec{Z}_{\omega_k}$ as $\vec{Z}_{\omega_k}^{a,b,c} = \cos(\omega_k(a + b - c))$. Since $\vec{Z}_{\omega_k} = \vec{Z}_{\omega_{p-k}}$, we set $1 \le k \le K$, where $K = \lceil (p-1)/2 \rceil$, to obtain $K$ distinct vectors, ignoring the constant bias vector. These vectors are orthogonal, as they are part of a Fourier basis.

Notice that any circuit that was exactly following the learned algorithm described above would only produce logits in the directions $\vec{Z}_{\omega_k}$ for the key frequencies $\omega_k$. So, we can define the trigonometric contribution to $\vec{Z}_f$ as the projection of $\vec{Z}_f$ onto the directions $\vec{Z}_{\omega_k}$. We may not know the key frequencies in advance, but we can sum over all $K$ of them, giving the following definition for trigonometric logits:

$$\vec{Z}_{f,T} = \sum_{k=1}^{K} (\vec{Z}_f \cdot \hat{Z}_{\omega_k}) \hat{Z}_{\omega_k}$$

where $\hat{Z}_{\omega_k}$ is the normalised version of $\vec{Z}_{\omega_k}$. This corresponds to projecting onto a $K$-dimensional subspace of the $p^3$-dimensional space in which $\vec{Z}_f$ lives.

**Memorisation logits.** Early in training, neural networks memorise the training dataset without generalising, suggesting that there exists a memorisation algorithm, implemented by the memorising circuit[2]. Unfortunately, we do not understand the algorithm underlying memorisation, and so cannot design a similar procedure to isolate the memorising circuit's contribution to the logits. However, we hypothesise that for modular addition, generalisation and memorisation are the only two circuit families of importance for the loss. This allows us to define the memorisation contribution to the logits as the residual:

$$\vec{Z}_{f,M} = \vec{Z}_f - \vec{Z}_{f,T}$$

**Trigonometric and memorisation circuits.** We say that a circuit is a trigonometric circuit if it implements the trigonometric algorithm, and similarly for memorisation circuits. Importantly, this is a many-to-one mapping: there are many possible circuits that implement a given algorithm.

We isolate trigonometric ($\vec{o}_t$) and memorisation ($\vec{o}_m$) logits by projecting the output logits ($\vec{o}$) as described in Appendix C. We cannot directly measure the circuit weights $w_t$ and $w_m$, but instead use an indirect measure: the value of the logit for the correct class given by each circuit, i.e. $o_t^y$ and $o_m^y$.

**Flaws** These metrics should be viewed as an imperfect proxy measure for the true strength of the trigonometric and memorisation circuits, as they have a number of flaws:

1. When both trigonometric and memorisation circuits are present in the network, they are both expected to produce high values for the correct logits, and low values for incorrect logits, on the train dataset. Since the trigonometric and memorisation logits are correlated, it becomes more likely that $\vec{Z}_{f,T}$ captures memorisation logits too.

2. In this case we would expect our proxy measure to overestimate the strength of the trigonometric circuit and underestimate the strength of memorisation. In fact, in our experiments we do see large negative correct logit values for memorisation on training for semi-grokking, which probably arises because of this effect.

3. Logits are not inherently meaningful; what matters for loss is the extent to which the correct logit is larger than the incorrect logits. This is not captured by our proxy metric, which only looks at the size of the correct logit. In a binary classification setting, we could instead use the difference between the correct and incorrect logit, but it is not clear what a better metric would be in the multiclass setting.

---

[2]In reality, there are at least two different memorisation algorithms: commutative memorisation (which predicts the same answer for $(a, b)$ and $(b, a)$) and non-commutative memorisation (which does not). However, this difference does not matter for our analyses, and we will call both of these "memorisation" in this paper.

# D  PROOFS OF THEOREMS

We assume we have a set of inputs $X$, a set of labels $Y$, and a training dataset, $\mathcal{D} = \{(x_1, y_1), \ldots (x_D, y_D)\}$. Let $f$ be a classifier that assigns a real-valued logit for each possible label given an input. We denote an individual logit as $o_f^y(x) := f(x, y)$. When the input $x$ is clear from context, we will denote the logit as $o_f^y$. Excluding weight decay, the *loss* for the classifier is given by the softmax cross-entropy loss:

$$\mathcal{L}_{\text{x-ent}}(f) = -\frac{1}{D} \sum_{(x,y) \in \mathcal{D}} \log \frac{\exp(o_f^y)}{\sum\limits_{y' \in Y} \exp(o_f^{y'})}.$$

For any $c \in \mathbb{R}$, let $c \cdot f$ be the classifier whose logits are multipled by $c$, that is, $(c \cdot f)(x, y) = c \times f(x, y)$. Intuitively, once a classifier achieves perfect accuracy, then the true class logit $o^{y^*}$ will be larger than any incorrect class logit $o^{y'}$, and so loss can be further reduced by scaling up *all* of the logits further (increasing the gap between $o^{y^*}$ and $o^{y'}$).

**Theorem D.1.** *Suppose that the classifier $f$ has perfect accuracy, that is, for any $(x, y^*) \in \mathcal{D}$ and any $y' \neq y^*$ we have $o_f^{y^*} > o_f^{y'}$. Then, for any $c > 1$, we have $\mathcal{L}_{\text{x-ent}}(c \cdot f) < \mathcal{L}_{\text{x-ent}}(f)$.*

*Proof.* First, note that we can rewrite the loss function as:

$$\mathcal{L}_{\text{x-ent}}(f) = -\frac{1}{D} \sum_{(x,y^*)} \log \frac{\exp(o_f^{y^*})}{\sum_{y'} \exp(o_f^{y'})} = \frac{1}{D} \sum_{(x,y^*)} \log \left( \frac{\sum_{y'} \exp(o_f^{y'})}{\exp(o_f^{y^*})} \right) = \frac{1}{D} \sum_{(x,y^*)} \log \left( 1 + \sum_{y' \neq y^*} \exp(o_f^{y'} - o_f^{y^*}) \right)$$

Since we are given that $o_f^{y^*} > o_f^{y'}$, for any $c > 1$ we have $c(o_f^{y'} - o_f^{y^*})) < o_f^{y'} - o_f^{y^*}$. Since exp, log, and sums are all monotonic, this gives us our desired result:

$$\mathcal{L}_{\text{x-ent}}(c \cdot f) = \frac{1}{D} \sum_{(x,y^*)} \log \left( 1 + \sum_{y' \neq y^*} \exp(c(o_f^{y'} - o_f^{y^*})) \right) < \frac{1}{D} \sum_{(x,y^*)} \log \left( 1 + \sum_{y' \neq y^*} \exp(o_f^{y'} - o_f^{y^*}) \right) = \mathcal{L}_{\text{x-ent}}(f).$$

$\square$

We now move on to Theorem D.4. First we establish some basic lemmas that will be used in the proof:

**Lemma D.2.** *Let $a, b, r \in \mathbb{R}$ with $a, b \geq 0$ and $0 < r \leq 1$. Then $(a + b)^r \leq a^r + b^r$.*

*Proof.* The case with $a = 0$ or $b = 0$ is clear, so let us consider $a, b > 0$. Let $x = \frac{a}{a+b}$ and $y = \frac{b}{a+b}$. Since $0 \leq x \leq 1$, we have $x^{(1-r)} \leq 1$, which implies $x \leq x^r$. Similarly $y \leq y^r$. Thus $x^r + y^r \geq x + y = 1$. Substituting in the values of $x$ and $y$ we get $\frac{a^r + b^r}{(a+b)^r} \geq 1$, which when rearranged gives us the desired result. $\square$

**Lemma D.3.** *For any $x, c, r \in \mathbb{R}$ with $r \geq 1$, there exists some $\delta > 0$ such that for any $\epsilon < \delta$ we have $x^r - (x - \epsilon)^r > \delta(rx^{r-1} - c)$.*

*Proof.* The function $f(x) = x^r$ is everywhere-differentiable and has derivative $rx^{r-1}$. Thus we can choose $\delta$ such that for any $\epsilon < \delta$ we have $-c < \frac{x^r - (x-\epsilon)^r}{\delta} - rx^{r-1} < c$. Rearranging, we get $x^r - (x - \epsilon)^r > \delta(rx^{r-1} - c)$ as desired. $\square$

### D.1 Weight decay favours efficient circuits

To flesh out the argument in Section 2, we construct a minimal example of multiple circuits $\{C_1, \ldots C_I\}$ of varying efficiencies that can be scaled up or down through a set of non-negative *weights* $w_i$. Our classifier is given by $f = \sum_{i=1}^{I} w_i C_i$, that is, the output $f(x, y)$ is given by $\sum_{i=1}^{I} w_i C_i(x, y)$.

We take circuits $C_i$ that are *normalised*, that is, they produce the same average logit value. $P_i$ denotes the parameter norm of the normalised circuit $C_i$. We decide to call a circuit with lower $P_i$ more efficient. However, it is hard to define efficiency precisely. Consider instead the parameter norm $P_i'$ of the scaled circuit $w_i C_i$. If we define efficiency as either the ratio $\|\vec{o}_{C_i}\|/P_i'$ or the derivative $d\|\vec{o}_{C_i}\|/dP_i'$, then it would vary with $w_i$ since $\vec{o}_{C_i}$ and $P_i'$ can in general have different relationships with $w_i$. We prefer $P_i$ as a measure of relative efficiency as it is intrinsic to $C_i$ rather than depending on its scaling $w_i$.

Gradient descent operates over the weights $w_i$ (but not $C_i$ or $P_i$) to minimise $\mathcal{L} = \mathcal{L}_{\text{x-ent}} + \alpha \mathcal{L}_{\text{wd}}$. $\mathcal{L}_{\text{x-ent}}$ can easily be rewritten in terms of $w_i$, but for $\mathcal{L}_{\text{wd}}$ we need to model the parameter norm of the scaled circuits $w_i C_i$. Notice that, in a $\kappa$-layer MLP with Relu activations and without biases, scaling all parameters by a constant $c$ scales the outputs by $c^\kappa$. Inspired by this observation, we model the parameter norm of $w_i C_i$ as $w_i^{1/\kappa} P_i$ for some $\kappa > 0$. This gives the following effective loss:

$$\mathcal{L}(\vec{w}) = \mathcal{L}_{\text{x-ent}} \left( \sum_{i=1}^{I} w_i C_i \right) + \frac{\alpha}{2} \sum_{i=1}^{I} (w_i^{\frac{1}{\kappa}} P_i)^2$$

We will generalise this to any $L_q$-norm (where $q > 0$). Standard weight decay corresponds to $q = 2$. We will also generalise to arbitrary differentiable, bounded training loss functions, instead of cross-entropy loss specifically. In particular, we assume that there is some differentiable $\mathcal{L}_{\text{train}}(f)$ such that there exists a finite bound $B \in \mathbb{R}$ such that $\forall f : \mathcal{L}_{\text{train}}(f) \geq B$. (In the case of cross-entropy loss, $B = 0$.)

With these generalisations, the overall loss is now given by:

$$\mathcal{L}(\vec{w}) = \mathcal{L}_{\text{train}} \left( \sum_{i=1}^{I} w_i C_i \right) + \frac{\alpha}{q} \sum_{i=1}^{I} (w_i^{\frac{1}{\kappa}} P_i)^q \tag{5}$$

The following theorem establishes that the optimal weight vector allocates more weight to more efficient circuits, under the assumption that the circuits produce identical logits on the training dataset.

**Theorem D.4.** *Given $I$ circuits $C_i$ and associated $L_q$ parameter norms $P_i$, assume that every circuit produces the same logits on the training dataset, i.e. $\forall i, j, \forall (x, \_) \in \mathcal{D}, \forall y' \in Y$ we have $o_{C_i}^{y'}(x) = o_{C_j}^{y'}(x)$. Then, any weight vector $\vec{w}^* \in \mathbb{R}^I$ that minimizes the loss in Equation 5 subject to $w_i \geq 0$ satisfies:*

    *1. If $\kappa \geq q$, then $w_i^* = 0$ for all $i$ such that $P_i > \min_j P_j$.*

    *2. If $0 < \kappa < q$, then $w_i^* \propto P_i^{-\frac{q\kappa}{q-\kappa}}$.*

*Intuition .* Since every circuit produces identical logits, their weights are interchangeable with each other from the perspective of $\mathcal{L}_{\text{x-ent}}$, and so we must analyse how interchanging weights affects $\mathcal{L}_{\text{wd}}$. $\mathcal{L}_{\text{wd}}$ grows as $O(w_i^{2/\kappa})$. When $\kappa > 2$, $\mathcal{L}_{\text{wd}}$ grows sublinearly, and so it is cheaper to add additional weight to the *largest* weight, creating a "rich get richer" effect that results in a single maximally efficient circuit getting all of the weight. When $\kappa < 2$, $\mathcal{L}_{\text{wd}}$ grows superlinearly, and so it is cheaper to add additional weight to the *smallest* weight. As a result, every circuit is allocated at least some weight, though more efficient circuits are still allocated higher weight than less efficient circuits.

*Sketch .* The assumption that every circuit produces the same logits on the training dataset implies that $\mathcal{L}_{\text{train}}$ is purely a function of $\sum_{i=1}^{I} w_i$. So, for $\mathcal{L}_{\text{train}}$, a small increase $\delta w$ to $w_i$ can be balanced by a corresponding decrease $\delta w$ to some other weight $w_j$.

For $\mathcal{L}_{\text{wd}}$, an increase $\delta w$ to $w_i$ produces a change of approximately $\frac{\delta \mathcal{L}_{\text{wd}}}{\delta w_i} \cdot \delta w = \frac{\alpha}{\kappa} \left(P_i(w_i)^r\right)^q \cdot \delta w$, where $r = \frac{1}{\kappa} - \frac{1}{q} = \frac{q-\kappa}{q\kappa}$. So, an increase of $\delta w$ to $w_i$ can be balanced by a decrease of $\left(\frac{P_i(w_i)^r}{P_j(w_j)^r}\right)^q \delta w$ to some other weight $w_j$. The two cases correspond to $r \leq 0$ and $r > 0$ respectively.

**Case 1:** $r \leq 0$. Consider $i, j$ with $P_j > P_i$. The optimal weights must satisfy $w_i^* \geq w_j^*$ (else you could swap $w_i^*$ and $w_j^*$ to decrease loss). But then $w_j^*$ must be zero: if not, we could increase $w_i^*$ by $\delta w$ and decrease $w_j^*$ by $\delta w$, which keeps $\mathcal{L}_{\text{x-ent}}$ constant and decreases $\mathcal{L}_{\text{wd}}$ (since $P_i(w_i^*)^r < P_j(w_j^*)^r$).

**Case 2:** $r > 0$. Consider $i, j$ with $P_j > P_i$. As before we must have $w_i^* \geq w_j^*$. But now $w_j^*$ must *not* be zero: otherwise we could increase $w_j^*$ by $\delta w$ and decrease $w_i^*$ by $\delta w$ to keep $\mathcal{L}_{\text{x-ent}}$ constant and decrease $\mathcal{L}_{\text{wd}}$, since $P_j(w_j^*)^r = 0 < P_i(w_i^*)^r$. The balance occurs when $P_j(w_j^*)^r = P_i(w_i^*)^r$, which means $w_i^* \propto P_i^{-1/r}$.

*Proof.* First, notice that our conclusions trivially hold for $\vec{w}^* = \vec{0}$ (which can be a minimum if e.g. the circuits are worse than random). Thus for the rest of the proof we will assume that at least one weight is non-zero.

In addition, $\mathcal{L} \to \infty$ whenever any $w_i \to \infty$ (because $\mathcal{L}_{\text{train}} \geq B$ and $\mathcal{L}_{\text{wd}} \to \infty$ as any one $w_i \to \infty$). Thus, any global minimum must have finite $\vec{w}$.

Notice that, since the circuit logits are independent of $i$, we have $f = (\sum_i w_i) f$, and so $\mathcal{L}_{\text{train}}(\vec{w})$ is purely a function of the sum of weights $\sum_{i=1}^{I} w_i$, and the overall loss can be written as:

$$\mathcal{L}(\vec{w}) = \mathcal{L}_{\text{train}}\left(\sum_{i=1}^{I} w_i\right) + \frac{\alpha}{q} \sum_{i=1}^{I} ((w_i)^{\frac{1}{\kappa}} P_i)^q$$

We will now consider each case in order.

**Case 1:** $\kappa \geq q$. Assume towards contradiction that there is a global minimum $\vec{w}^*$ where $w_j^* > 0$ for some circuit $C_j$ with non-minimal $P_j$. Let $C_i$ be a circuit with minimal $P_i$ (so that $P_i < P_j$), and let its weight be $w_i^*$.

Consider an alternate weight assignment $\vec{w}'$ that is identical to $\vec{w}^*$ except that $w_j' = 0$ and $w_i' = w_i^* + w_j^*$. Clearly $\sum_i w_i^* = \sum_i w_i'$, and so $\mathcal{L}_{\text{train}}(\vec{w}^*) = \mathcal{L}_{\text{train}}(\vec{w}')$. Thus, we have:

$\mathcal{L}(\vec{w}^*) - \mathcal{L}(\vec{w}')$

$= \left(\frac{\alpha}{q} \sum_{m=1}^{I} ((w_m^*)^{\frac{1}{\kappa}} P_m)^q\right) - \left(\frac{\alpha}{q} \sum_{m=1}^{I} ((w_m')^{\frac{1}{\kappa}} P_m)^q\right)$

$= \frac{\alpha}{q} \left((w_i^*)^{\frac{q}{\kappa}} P_i^q + (w_j^*)^{\frac{q}{\kappa}} P_j^q - (w_i')^{\frac{q}{\kappa}} P_i^q\right)$

$> \frac{\alpha}{q} P_i^q \left((w_i^*)^{\frac{q}{\kappa}} + (w_j^*)^{\frac{q}{\kappa}} - (w_i')^{\frac{q}{\kappa}}\right)$          since $P_j > P_i$

$= \frac{\alpha}{q} P_i^q \left((w_i^*)^{\frac{q}{\kappa}} + (w_j^*)^{\frac{q}{\kappa}} - (w_i^* + w_j^*)^{\frac{q}{\kappa}}\right)$          definition of $w_i'$

$\geq \frac{\alpha}{q} P_i^q \left((w_i^*)^{\frac{q}{\kappa}} + (w_j^*)^{\frac{q}{\kappa}} - \left((w_i^*)^{\frac{q}{\kappa}} + (w_j^*)^{\frac{q}{\kappa}}\right)\right)$     using Lemma D.2 since $0 < \frac{q}{\kappa} \leq 1$

$= 0$

Thus we have $\mathcal{L}(\vec{w}^*) > \mathcal{L}(\vec{w}')$, contradicting our assumption that $\vec{w}^*$ is a global minimum of $\mathcal{L}$. This completes the proof for the case that $\kappa \geq q$.

**Case 2:** $\kappa < q$. First, we will show that all weights are non-zero at a global minimum (excluding the case where $\vec{w}^* = \vec{0}$, discussed at the beginning of the proof). Assume towards contradiction that there is a global minimum $\vec{w}^*$ with $w_j^* = 0$ for some $j$. Choose some arbitrary circuit $C_i$ with nonzero weight $w_i^*$.

Choose some $\epsilon_1 > 0$ satisfying $\epsilon_1 < \frac{q}{2\kappa}(w_i^*)^{\frac{q}{\kappa}-1}$. By applying Lemma D.3 with $x = w_i^*, c = \epsilon_1, r = \frac{q}{\kappa}$, we can get some $\delta > 0$ such that for any $\epsilon < \delta$ we have $(w_i^*)^{\frac{q}{\kappa}} - (w_i^* - \epsilon)^{\frac{q}{\kappa}} > \delta(\frac{q}{\kappa}(w_i^*)^{\frac{q}{\kappa}-1} - \epsilon_1)$.

Choose some $\epsilon_2 > 0$ satisfying $\epsilon_2 < \min(w_i^*, \delta, \left[\frac{q}{2\kappa}(w_i^*)^{\frac{q}{\kappa}-1}\frac{P_i^q}{P_j^q}\right]^{\frac{1}{\frac{q}{\kappa}-1}})$. Consider an alternate weight assignment defined $\vec{w}'$ that is identical to $\vec{w}^*$ except that $w_j' = \epsilon_2$ and $w_i' = w_i^* - \epsilon_2$. As in the previous case, $\mathcal{L}_{\text{train}}(\vec{w}^*) = \mathcal{L}_{\text{train}}(\vec{w}')$. Thus, we have:

$$\mathcal{L}(\vec{w}^*) - \mathcal{L}(\vec{w}')$$

$$= \frac{\alpha}{q}\left((w_i^*)^{\frac{q}{\kappa}}P_i^q - (w_i^* - \epsilon_2)^{\frac{q}{\kappa}}P_i^q - \epsilon_2^{\frac{q}{\kappa}}P_j^q\right)$$

$$= \frac{\alpha}{q}\left(P_i^q((w_i^*)^{\frac{q}{\kappa}} - (w_i^* - \epsilon_2)^{\frac{q}{\kappa}}) - \epsilon_2^{\frac{q}{\kappa}}P_j^q\right)$$

$$> \frac{\alpha}{q}\left(P_i^q\delta(\frac{q}{\kappa}(w_i^*)^{\frac{q}{\kappa}-1} - \epsilon_1) - \epsilon_2^{\frac{q}{\kappa}}P_j^q\right) \qquad \text{application of Lemma D.3 discussed above}$$

$$> \frac{\alpha}{q}\left(P_i^q\delta(\frac{q}{\kappa}(w_i^*)^{\frac{q}{\kappa}-1} - \frac{q}{2\kappa}(w_i^*)^{\frac{q}{\kappa}-1}) - \epsilon_2^{\frac{q}{\kappa}}P_j^q\right) \qquad \text{we chose } \epsilon_1 < \frac{q}{2\kappa}(w_i^*)^{\frac{q}{\kappa}-1}$$

$$> \frac{\alpha}{q}\left(P_i^q\epsilon_2\frac{q}{2\kappa}(w_i^*)^{\frac{q}{\kappa}-1} - \epsilon_2^{\frac{q}{\kappa}}P_j^q\right) \qquad \text{we chose } \epsilon_2 < \delta$$

$$= \frac{\alpha\epsilon_2}{q}\left(\frac{q}{2\kappa}(w_i^*)^{\frac{q}{\kappa}-1}P_i^q - \epsilon_2^{\frac{q}{\kappa}-1}P_j^q\right)$$

$$> \frac{\alpha\epsilon_2}{q}\left(\frac{q}{2\kappa}(w_i^*)^{\frac{q}{\kappa}-1}P_i^q - \frac{q}{2\kappa}(w_i^*)^{\frac{q}{\kappa}-1}\frac{P_i^q}{P_j^q}P_j^q\right) \qquad \text{we chose } \epsilon_2 < \left[\frac{q}{2\kappa}(w_i^*)^{\frac{q}{\kappa}-1}\frac{P_i^q}{P_j^q}\right]^{\frac{1}{\frac{q}{\kappa}-1}})$$

$$= 0$$

Note that in the last step, we rely on the fact that $\kappa < q$: this lets us use an upper bound on $\epsilon_2$ to get an upper bound on $\epsilon_2^{\frac{q}{\kappa}-1}$, and so a lower bound on the overall expression.

Thus we have $\mathcal{L}(\vec{w}^*) > \mathcal{L}(\vec{w}')$, contradicting our assumption that $\vec{w}^*$ is a global minimum of $\mathcal{L}$. So, for all $i$ we have $w_i > 0$.

In addition, as $w_i \to \infty$ we have $\mathcal{L}(\vec{w}) \to \infty$, so $\vec{w}^*$ cannot be at the boundaries, and instead lies in the interior. Since $q > \kappa$, $\mathcal{L}(\vec{w})$ is differentiable everywhere. Thus, we can conclude that its gradient at $\vec{w}^*$ is zero:

$$\frac{\delta\mathcal{L}}{\delta w_i} = 0$$

$$\frac{\delta\mathcal{L}_{\text{train}}}{\delta w_i} + \frac{\alpha P_i^q}{\kappa}(w_i^*)^{\frac{q}{\kappa}-1} = 0$$

$$P_i^q(w_i^*)^{\frac{q-\kappa}{\kappa}} = -\frac{\kappa}{\alpha}\frac{\delta\mathcal{L}_{\text{train}}}{\delta w_i}$$

$$w_i^*P_i^{\frac{q\kappa}{q-\kappa}} = \left(-\frac{\kappa}{\alpha}\frac{\delta\mathcal{L}_{\text{train}}}{\delta w_i}\right)^{\frac{\kappa}{q-\kappa}}$$

Since $\mathcal{L}_{\text{train}}(\vec{w})$ is a function of $\sum_{j=1}^{I} w_j$, we can conclude that $\frac{\delta\mathcal{L}_{\text{train}}}{\delta w_i} = \frac{\delta\mathcal{L}_{\text{train}}}{\delta\sum_j w_j} \cdot \frac{\delta\sum_j w_j}{\delta w_i} = \frac{\delta\mathcal{L}_{\text{train}}}{\delta\sum_j w_j}$, which is independent of $i$. So the right hand side of the equation is independent of $i$, allowing us to conclude that $w_i^* \propto P_i^{-\frac{q\kappa}{q-\kappa}}$.

$\square$

