# OpenReview forum: "Explaining grokking through circuit efficiency"
_ICLR.cc/2024/Conference — Submitted to ICLR 2024_

### Official Review · Reviewer_m2MB · 2023-10-28

**Soundness:** 3 good
**Presentation:** 4 excellent
**Contribution:** 3 good
**Rating:** 6
**Confidence:** 4

**Summary:**

The paper presents a theory of grokking in neural networks based on the concept of sub-networks. Based on the theory, many interesting behaviors are predicted and verified. The paper is well written and the theory is novel to me. The only issue that prevents me from giving a higher score is the relatively simple setting of the problem (see weakness part). However, I believe the current version is good enough for ICLR.

**Strengths:**

1. I believe the “ungrokking” and “semi-grokking” phenomenons are quite persuasive: continuing training on a subset after the grokking point makes the test loss drop again; training on a dataset of a specific size will make the training loss fluctuate around a very low level, and the test loss will hover around some intermediate value. It makes me believe that the two types of circuits are “competing” during the learning, and the size of the training set decides which will win. The results in Figure 4 also demonstrate that weight decay doesn’t influence the value of $D_{crit}$.
2. The paper is well-written and quite easy to follow.

**Weaknesses:**

1. How do the findings in grokking help us understand emergent behavior better? How could these results guide the design of our deep learning systems? It would be nice if the paper included some discussion about this.
2. Although most of the papers in this direction consider a similar setting that contains only x and y as input, is it possible to extend the analysis to a more general setting, e.g.,$(a+b+c*d)$ mod $p$?
3. Similar concerns for the network structure. Will the analysis (or even the grokking phenomenon) still work for non-transformer models?
4. Similar concerns for the way we encode the values of the input. Will the analysis (or even the grokking phenomenon) still work when the input signal is encoded in different ways?

**Questions:**

1. I think using “norm of parameters” to measure efficiency is not good, as we can create the same function by multiplying c to one layer and multiplying 1/c to another layer: the same function can have different parameter norms. I speculate the concept of efficiency is related to “how complex the function is”. Memorizing a circuit would be quite complex as it cannot uncover the ground truth rules and have to remember everything, generalizing a circuit would be simpler as it captures the rules. IMO, these two concepts are quite similar to the holistic mapping and compositional mapping mentioned in [1], which evaluate the generalization ability of different mappings using coding length. In summary, rather than the parameter’s norm, I suggest considering coding length, Kolmogorov complexity, or even sample efficiency (number of training samples needed to make the model generalize well) to compare how efficient a circuit is.
2. Why do memorizing circuits learn faster than generalizing circuits in the setting of grokking? This counters with my intuition, because we usually believe memorizing happens in the overfitting phase, which is the latter phase of training. What is the difference between the settings of general supervised learning and grokking?
3. Are there any methods that can probe the model and allow us to directly observe these circuits? As discussed in the strength part, although the semi-grokking and the ungrokking behavior are strong evidence of the proposed explanation, some other mechanisms might also cause similar behavior. For example, ungrokking might be caused by catastrophic forgetting: the subset used for continuous training might contain some poison samples that harm the generalization ability. So I believe more evidence would make the paper’s claim more solid.

[1] Yi Ren, Samuel Lavoie, et. al. Improving Compositional Generalization using Iterated Learning and Simplicial Embeddings, NeurIPS 2023

---

> ### Author Response · Authors · 2023-11-19
>
> Thanks for the review! We’re glad to see that you were convinced of our explanation, particularly through the demonstrations of ungrokking and semi-grokking.
>
> ## Applicability
>
> > How do the findings in grokking help us understand emergent behavior better? How could these results guide the design of our deep learning systems?
>
> Unfortunately, we aren’t aware of any immediate applications of these findings (which we allude to in Section 6). In particular, we expect that typical cases of emergent phenomena (e.g. new capabilities with increased scale in LLMs) are better explained by the structure in the training dataset, rather than by the three ingredients we identify.
>
> However, the history of science suggests that it is often valuable to investigate and explain confusing phenomena, and that the resulting insights can have value that is hard to see in advance. As a result we are optimistic about contributing to our general understanding of deep learning systems, with the hope that it will guide our design of deep learning systems in the future.
>
> ## Speed of learning for generalisation
>
> > Why do memorizing circuits learn faster than generalizing circuits in the setting of grokking? This counters with my intuition, because we usually believe memorizing happens in the overfitting phase, which is the latter phase of training. What is the difference between the settings of general supervised learning and grokking?
>
> We already know that memorisation circuits can be learned reasonably quickly for a variety of architectures and input encodings – since training on random labels usually works quite well. We expect that in the algorithmic settings where we only have two families of circuits, the generalising circuit will be composed of multiple parts that all have to be present to provide any signal – in such situations we should expect the circuit to be learned slowly (see Appendix B5 and Jermyn & Shlegeris (2022)).
>
> In contrast, in general supervised learning, there are many simple statistical patterns whose circuits probably do not involve multiple parts (e.g. n-gram maps in language modeling), and so these can be learned quickly and generalise well.
>
> ## Generalisation to other settings
>
> > is it possible to extend the analysis to a more general setting, e.g., $(a+b+c∗d) \mod p$?
>
> > Similar concerns for the network structure. Will the analysis (or even the grokking phenomenon) still work for non-transformer models?
>
> > Similar concerns for the way we encode the values of the input. Will the analysis (or even the grokking phenomenon) still work when the input signal is encoded in different ways?
>
> We expect that the grokking phenomenon will probably persist in these settings, assuming one trains for long enough. (Note though that it may require much longer training than in the original setting, e.g. if the generalisation algorithm becomes harder to learn.) Conditional on grokking persisting, we expect that the analysis will still work.
>
> For all of these changes to the setting, the main consideration is how it affects the set of circuits that can solve the task. In particular, we need to know whether it remains the case that there are only two families of circuits (full generalisation and pure memorisation) that can solve the task.
>
> If the answer is no, then grokking will not persist (as that was one of the three crucial ingredients), and obviously our analysis will not apply.
>
> If the answer is yes, then we think the other two ingredients will also transfer:
> 1. Efficiency: As argued in Section 3.1, we expect that at sufficiently large dataset sizes, generalisation will be more efficient than memorisation, independent of the task structure, architecture, or input encoding. The critical dataset size may change, of course – we are just saying that there will still be _some_ critical dataset size.
> 2. Speed of learning: The argument above about speed of learning is independent of task structure, architecture, and input encoding – it just depends on the generalising circuit having multiple components that are all required to function well, which we expect will continue to hold in the algorithmic tasks where grokking has already been shown.
>
> Thus we’d also expect that grokking would persist, and the analysis will continue to apply.

---

> ### Author Response · Authors · 2023-11-19
> **Continuing our response**
>
> ## Miscellaneous questions
>
> > I think using “norm of parameters” to measure efficiency is not good, as we can create the same function by multiplying c to one layer and multiplying 1/c to another layer
>
> While this is the same function, it does _not_ achieve the same loss, because it changes the L2 norm of the parameters (and we are using weight decay, which to a first approximation penalises L2 norm of the parameters).
>
> > I speculate the concept of efficiency is related to “how complex the function is”. [...] I suggest considering coding length, Kolmogorov complexity, or even sample efficiency (number of training samples needed to make the model generalise well) to compare how efficient a circuit is.
>
> We certainly agree that efficiency is related to complexity. However, gradient descent does not find the best function by the metric of coding length / Kolmogorov complexity – as a simple example, if the shortest program takes a very long time to compute, then it cannot be implemented by the Transformer architecture.
>
> In general we think the appropriate notion of efficiency will depend on the particular training setup. We’d recommend (1) identifying the source of regularisation towards simplicity / low complexity, and (2) creating a metric that captures what the regularisation is optimising for, and using that as an operationalisation of efficiency.
>
> In our setting, (1) the source of regularisation is weight decay, and (2) the metric it optimises is the L2 norm of the parameters, which is why we consider parameter norm as the appropriate metric to quantify efficiency. In a setting without weight decay, we would want a metric that quantifies the implicit regularization of SGD (though unfortunately we are not sure how to create such a metric).
>
> (As a minor note, sample efficiency would clearly not be the right metric, because memorization sample efficiency is clearly lower than generalisation sample efficiency.)
>
> > Are there any methods that can probe the model and allow us to directly observe these circuits?
>
> Yes – as we describe at the beginning of Section 4, Nanda et al (2023) produce compelling evidence of these circuits, and create progress measures that show that the generalising circuit is learned slowly.
>
> > ungrokking might be caused by catastrophic forgetting: the subset used for continuous training might contain some poison samples that harm the generalization ability.
>
> This is not a plausible hypothesis:
> 1. This hypothesis doesn’t explain the phase transition we observe in ungrokking (see Figure 4).
> 2. If the problem were poison samples that harm generalisation, then randomly sampled large datasets should be at least as bad as randomly sampled small datasets, as the large datasets would be at least as likely to contain the poison samples. But we observe the opposite in Figure 4.

---

> > ### Comment · Reviewer_m2MB · 2023-11-22
> >
> > Thanks very much for the authors' response. Most of my concerns are addressed, but I still have something unclear. For the learning speed part, the authors claim that "... since training on random labels usually works quite well..." It is true that the model can learn noisy labels with 100% training accuracy. But from Figure 1 of [1], we see the speed of convergence of learning the true label is much larger than learning from the noisy label, which is a sign that the generalizing function (the one that aligns with the true data distribution) is learned faster than the memorizing function. Also, the authors mentioned that "... memorization sample efficiency is clearly lower than generalisation sample efficiency." IMO, if this is true, then the memorization circuit should be learned slower than the generalization one, because of its low sample efficiency. Furthermore, since sample efficiency can distinguish these two types of circuit, why it is not a good metric?
> >
> > [1] Zhang, Chiyuan, et al. "Understanding deep learning (still) requires rethinking generalization." Communications of the ACM 64.3 (2021): 107-115.

---

> ### Author Response · Authors · 2023-11-22
> **Generalisation is only slow for algorithmic tasks that show grokking**
>
> >  But from Figure 1 of [1], we see the speed of convergence of learning the true label is much larger than learning from the noisy label, which is a sign that the generalizing function (the one that aligns with the true data distribution) is learned faster than the memorizing function.
>
> Yes, for most settings we expect that the generalising function will be learned faster than the memorising one -- we think that the algorithmic tasks on which grokking occurs are _unusual_ in that their generalising functions are learned slowly rather than quickly. This is why grokking only happens for these algorithmic tasks, but doesn't usually happen in typical settings.
>
> See also our response to reviewer A2EW, under "First ingredient".
>
> > IMO, if this is true, then the memorization circuit should be learned slower than the generalization one, because of its low sample efficiency.
>
> Sorry, we thought that by "sample efficiency" you meant "how small does the dataset have to be to learn the circuit". If you instead mean "how many gradient updates do you have to take to learn the circuit", then it is true that the memorisation sample efficiency is higher than the generalisation sample efficiency (indeed sample efficiency is now a synonym for speed of learning).
>
> > Furthermore, since sample efficiency can distinguish these two types of circuit, why it is not a good metric?
>
> It may be a good metric in general -- we are just saying that it is not a good metric to operationalise our notion of "efficiency", because our notion of efficiency is meant to identify the strength of circuits at convergence, and to explain why (for large enough dataset sizes) generalisation is favoured over memorisation at convergence.

---

### Official Review · Reviewer_JVYB · 2023-11-01

**Soundness:** 3 good
**Presentation:** 4 excellent
**Contribution:** 2 fair
**Rating:** 3
**Confidence:** 5

**Summary:**

Grokking is the phenomenon by which models generalize long after overfitting. The paper aims to explain the phenomenon from a circuit efficiency perspective with a postulate that there are competing subnetworks: a generalizing subnetwork that is slow but more efficient than a memorizing subnetwork, which is fast but requires high complexity to accommodate a large training sample. The authors argue that these properties can explain delayed generalization.

**Strengths:**

- The writing is clear and well-structured and the authors laid out an interesting story explaining grokking.
- The paper tackles a very interesting phenomenon that can shed light on the dynamics of representation learning.
- The experiments are clean, and the visualizations are informative.

**Weaknesses:**

- On the empirical side, there are few results beyond modular arithmetic.
- On the theoretical side, there is a focus on the phenomenological observation that generalizing circuits are learned at a different speed compared to memorizing circuits, but the theory offers no explanation as to why they are slow in the first place. I think the real question is not whether or not the generalizing circuit is slower but rather *why* it is slower in this particular way, i.e., why does the model generalize so long after it overfits its training data? The origin of dynamics is crucial here.
- Furthermore, there is a heavy reliance on weight decay as an explanation for why generalizing circuits are favored, even though Grokking is known to occur without it. This is acknowledged in the paper but not addressed adequately.
- The efficiency metric based on parameter norm appears to have little to do with the main predictions and empirical observations (dataset size and semi-grokking and ungrokking). One can reach the same conclusions using only the simple premise that larger datasets are generally conducive to generalization while small ones are not (assuming consistent quality). Presumably, the transition between the two regimes depends on the specifics of the task.
- Many of the prior works cited studied the dependence of Grokking on data set size extensively. The novelty here is limited.
Overall, it's not clear this paper provides deeper insights than what is already in the literature so I cannot recommend acceptance.

**Questions:**

- Does the ungrokking setting maintain performance on the excluded part of the initial training set?
- It's hard to see how ungrokking is not a special instance of catastrophic forgetting (CF). I don't think the discussion making this distinction on page 5 is correct. Clearly, taking a specific subset of a dataset can be viewed as taking a different one, e.g., removing all points above a threshold effectively changes the training set distribution. The distinction from CF based on the choice of the (post)training set seems arbitrary. I would recommend dropping it.

---

> ### Author Response · Authors · 2023-11-19
>
> Thanks for the review! We are glad to see that you found the experiments clean and informative,  the paper easy to understand, and the topic studied interesting and significant.
>
> ## Novelty
>
> Our understanding is that your main critique of the paper is that the detailed principles we use are unnecessary and as a result the novelty is limited. However, we think that these detailed principles are in fact crucial to many of our predictions and experimental results.
>
> > One can reach the same conclusions using only the simple premise that larger datasets are generally conducive to generalization while small ones are not (assuming consistent quality).
>
> We disagree – our predictions do not follow just from this premise. In Section 4, we name the following four predictions:
>
> 1. (P1) Efficiency: We confirm our prediction that the generalising circuit efficiencies are independent of dataset size, while the memorising circuit efficiencies decrease as training dataset size increases.
> 2. (P2) Ungrokking (phase transition): We confirm our prediction that ungrokking shows a phase
> transition around $D_{\text{crit}}$.
> 3. (P3) Ungrokking (weight decay): We confirm our prediction that the final test accuracy after
> ungrokking is independent of the strength of weight decay.
> 4. (P4) Semi-grokking: We demonstrate that semi-grokking occurs in practice.
>
> We don’t think any of these predictions follow just from the premise that larger datasets are conducive to generalization while small ones are not:
>
> 1. P1 requires the notion of efficiency to state, and so is obviously specific to our theory.
> 2. P2 and P3 are about the shape of the curves in Figure 4, namely that there will be a phase transition (at a point that can be roughly identified from P1) and the shape of the curve will be independent of weight decay. Neither of these can be predicted from the general postulate you mention, and indeed we do not expect them to hold in typical settings.
> 3. P4 demonstrates a novel training curve where we see delayed generalization from ~zero test accuracy to middling test accuracy, which to our knowledge has not been predicted (or demonstrated) before.
>
> Inspired by your review, we demonstrate the points about P2 and P4 experimentally by running the same experimental procedure with MNIST instead of modular arithmetic, and find that indeed P2 and P4 do *not* apply in that setting (see Figures 7 and 10 in the updated paper).
>
> > Many of the prior works cited studied the dependence of Grokking on data set size extensively. The novelty here is limited.
>
> This is like saying “we already know that objects fall down, so there is no novelty in developing the theory of gravity”. Of course many other prior works have _observed_ the dependence of grokking on dataset size: our contribution is in _explaining_ why this dependence exists with a theory that also makes other novel predictions.
>
> Quoting from the introduction, our primary contributions are:
>
> 1. We explain how three ingredients can cause grokking (Section 2).
> 2. We explain why generalising circuits can be more efficient than memorising (Section 3.1).
> 3. Our theory implies a “critical dataset size” which we use to predict two novel phenomena:
> semi-grokking and ungrokking (Section 3.2).
> 4. We confirm our predictions empirically, providing support for the theory (Section 4).
>
> Some prior work vaguely gestures at (1) and (2), though not in as much detail and clarity as in our paper, and to our knowledge (3) and (4) are completely novel. (We agree that prior work established that there is a critical dataset size – our claim to novelty is in having a theory that predicts the critical dataset size, and the accompanying predictions of semi-grokking and ungrokking.)

---

> ### Author Response · Authors · 2023-11-19
> **Continuing our response**
>
> ## Other points
>
> > On the empirical side, there are few results beyond modular arithmetic.
>
> As a minor correction, we also have results on the symmetric group tasks from Power et al (see Figure 9).
>
> However, we assume that the reviewer’s critique is that we didn’t provide results in more realistic settings. This was a deliberate choice on our part, because we set out to explain why grokking happens. An important fact about grokking is that it does _not_ happen in realistic settings, and so a good explanation of grokking should _not_ apply to realistic settings. So, it doesn’t make sense to run experiments in realistic settings, except potentially to show that our explanation doesn’t apply to realistic settings (as in the MNIST experiment described above).
>
> We discuss this briefly in the Discussion as well.
>
> > I think the real question is not whether or not the generalizing circuit is slower but rather why it is slower in this particular way, i.e., why does the model generalize so long after it overfits its training data?
>
> We agree that this is an interesting question and see it as a natural avenue for future work. However, note that it was not a priori obvious that the three ingredients for grokking that we identify were in fact the necessary three ingredients – many other incompatible explanations for grokking have been proposed in the literature (such as slingshots, random walks in parameter space, high parameter norm at initialization, etc). We ask that the reviewer not diminish the contributions that we do provide, simply because they do not fully answer every question that we care about.
>
> Incidentally, we do have a hypothesis about why the generalizing circuit is slower, inspired by Jermyn & Shlegeris (2022). Specifically, a generalizing circuit is likely to be composed of many sequential operations / subcircuits, such that all of the operations have to be performed in order to achieve any reduction in loss. Concretely, we can model this as the generalizing circuit being compose of subcircuits A and B, where the overall effectiveness of the circuit is _multiplicative_ in terms of the strength of each of A and B. If we consider gradient descent on the function $A \cdot B$ with respect to $A$ and $B$, it is clear that the gradient for $A$ is proportional to the value of $B$ and vice versa – so if both $A$ and $B$ start out low (at random initialization, these circuits are only barely present), the gradients towards them are also very low, leading to very slow learning. This inspires our toy example, illustrated in Figure 2 and described in more detail in Appendix B.5. However, we still view this hypothesis as speculative, so we did not highlight it in the main body of the paper.
>
> > Furthermore, there is a heavy reliance on weight decay as an explanation for why generalizing circuits are favored, even though Grokking is known to occur without it. This is acknowledged in the paper but not addressed adequately.
>
> As we wrote in the paper (Section 6), any source of regularisation that favours generalisation can replace the role of weight decay in our theory. Research has already shown implicit regularisation from SGD, so given this there is a natural generalisation of our theory that explains grokking even without weight decay. What is unsatisfactory about this explanation?
>
> > Does the ungrokking setting maintain performance on the excluded part of the initial training set?
>
> No – the excluded part of the initial training set is included in the new test set, and for small enough dataset sizes the new test accuracy drops to effectively zero.
>
> > It's hard to see how ungrokking is not a special instance of catastrophic forgetting (CF). [...] Clearly, taking a specific subset of a dataset can be viewed as taking a different one, e.g., removing all points above a threshold effectively changes the training set distribution.
>
> We certainly agree that ungrokking involves changing the training set distribution. When we talked about a “new” dataset, we meant to imply that there are new data points in the train set that were not present in the previous train set – we will clarify that sentence to make this clear.
>
> On the broader point, whether or not ungrokking is a special instance of CF seems to depend on the definition of CF (e.g. whether it requires new data points not previously present or not). We don’t mean to make any claims about this, given that there isn’t an agreed upon, formal definition of CF (to our knowledge). Our claim is simply that ungrokking is quite different from _typical examples_ of CF, and isn’t something that we would have expected independently of our theory. For example, CF does not predict a critical size at which forgetting suddenly happens, whereas our theory does.

---

> > ### Comment · Reviewer_JVYB · 2023-11-23
> >
> > Thank you for the reply. Unfortunately, I do not think that my concerns about the novelty of the submission were addressed properly. Regarding the predictions being made:
> >
> > > (P1) Efficiency: We confirm our prediction that the generalising circuit efficiencies are independent of dataset size, while the memorising circuit efficiencies decrease as training dataset size increases.
> >
> > Just to clarify, the axiom proposed is that a generalizing circuit has a lower weight norm than a memorizing circuit. Then, from this axiom follows the prediction that the weight norm is fixed for a neural network that generalizes but not for one trained on random labels. My problem with this line of reasoning is that it is both broad and irrelevant at the same time. It is unclear to me how this relates to the grokking setting. The empirical evidence for the above is not shown on the original task, through understanding the memorizing circuit for example, but on a dataset with random labels. In this setup, we can immediately expect solutions to have a large (and increasing) weight norm. This can be simply due to the fact that while we generally expect “natural data” to lie on a well-behaved manifold, random labels induce a larger Lipschitz constant on the function we aim to approximate, which in turn leads to a higher weight norm. Thus, studying the weight norm in this setting introduces an important confounder and cannot be taken as evidence of the claims being made without proper treatment.
> >
> > (P2) Ungrokking is the catastrophic forgetting of the original training set when the model is further trained on a small subset. One would expect this by default, and can hardly be considered a novel prediction.
> >
> > > This is like saying “we already know that objects fall down, so there is no novelty in developing the theory of gravity”. Of course many other prior works have observed the dependence of grokking on dataset size: our contribution is in explaining why this dependence exists with a theory that also makes other novel predictions.
> >
> > My comment was that prior works extensively studied the dependence on train set size. E.g., upon inspecting Liu et al [1]., one can find that they explain mechanistically why there is a critical set size on the addition task by studying the parallelograms induced by the samples available in training. Not only does this predict the existence of a critical set size, but it also predicts a numerical value for the fraction of the data needed for generalization. Thus, the statement that this is a novel contribution is incorrect. That is not to say there is no room for other explanations, but the one presented in this paper is incomplete, and its predictions are certainly not novel.
> >
> > > P4 demonstrates a novel training curve where we see delayed generalization from ~zero test accuracy to middling test accuracy, which to our knowledge has not been predicted (or demonstrated) before.
> >
> > Was this not demonstrated in [1] (e.g., Figure 3)? They explain this observation by counting the number of parallelograms in the training set.
> >
> > > No – the excluded part of the initial training set is included in the new test set, and for small enough dataset sizes the new test accuracy drops to effectively zero.
> >
> > Then, I don’t understand the point being made in the paper. There is an explicit claim that ungrokking differs from CF because there is “only bad test loss.” But since the previous train set is now contained in the test set, for which performance degrades to effectively 0, it is necessarily implied that the performance on the old train set has degraded.
> >
> > [1] Towards Understanding Grokking: An Effective Theory of Representation Learning, Liu et al., 2022

---

> ### Author Response · Authors · 2023-11-23
> **Liu et al's explanation is different and does not apply to our setting**
>
> ## Liu et al
>
> We hadn't realized that you were thinking of Liu et al in your original comments -- we agree that Liu et al is presenting explanations and making predictions, not just observations, but we believe that these explanations and predictions do not apply to the typical grokking settings in modular arithmetic.
>
> For example, Liu et al assume:
> 1. Given an input "$a + b$", the embeddings for $a$ and $b$ are _added_, and only then is further processing done. This is a major architectural difference from typical settings. (For example, it bakes in the assumption that the task is symmetric.)
> 2. They study regular (non-modular) arithmetic instead of modular arithmetic, which makes the parallelogram representation useful.
>
> > upon inspecting Liu et al [1]., one can find that they explain mechanistically why there is a critical set size on the addition task by studying the parallelograms induced by the samples available in training.
>
> Liu et al's explanation is extremely different from ours: they make a primarily _information-theoretic_ argument (the information needed to generalise, i.e. parallelograms, have to be present in the training data to constrain the embeddings), whereas our explanation is tied to training dynamics and loss functions.
>
> > Not only does this predict the existence of a critical set size, but it also predicts a numerical value for the fraction of the data needed for generalization.
>
> Yes, but we expect the prediction would be incorrect if applied to our tasks. We are not sure how to use Liu et al to make predictions on our task given the major disconnects between their assumptions and our task setting, but if you have a suggestion for the methodology we can see whether it produces the right prediction in practice.
>
> > Was this not demonstrated in [1] (e.g., Figure 3)? They explain this observation by counting the number of parallelograms in the training set.
>
> Fair point -- we should say that it hasn't been predicted or demonstrated in a more typical setting (our setting is standard deep learning apart from the task, whereas the one in Liu et al is artificial in a variety of ways, particularly the architecture).
>
> ## Other points
>
> > Just to clarify, the axiom proposed is that a generalizing circuit has a lower weight norm than a memorizing circuit.
>
> This is when holding the scale of the logits fixed, but otherwise yes. (Weight norm of any individual circuit varies with the strength of the logits it produces.)
>
> > This can be simply due to the fact that while we generally expect “natural data” to lie on a well-behaved manifold, random labels induce a larger Lipschitz constant on the function we aim to approximate, which in turn leads to a higher weight norm. Thus, studying the weight norm in this setting introduces an important confounder and cannot be taken as evidence of the claims being made without proper treatment.
>
> While it is true that we expect there to be structure in natural data, e.g. lying on a well-behaved manifold, a memorising circuit does not take advantage of this structure -- if it did, then we would expect it to generalise at least partially. So we think this is a good way of measuring the quantities we care about. In addition, it's noteworthy that this empirical evidence (shown in Figure 3) suggests a critical dataset size that is roughly in line with the estimate we would get from ungrokking (shown in Figure 4).
>
> That said, we agree that this is not the most compelling evidence we provide, because of the potential for confounders given that our measurements are not on the original task.
>
> > Ungrokking is the catastrophic forgetting of the original training set when the model is further trained on a small subset. One would expect this by default, and can hardly be considered a novel prediction.
>
> We agree that it is not surprising that accuracy falls when the model is further trained on a small subset. As we noted in our rebuttal and as stated in both the original and updated paper, our novel prediction is specifically about the shape of the curve for ungrokking (namely that it will have a phase transition). Empirically this is borne out for grokking tasks, but does _not_ hold for regular tasks like MNIST (see Figure 10). If you "expected by default" a phase transition in test accuracy even for MNIST, your expectation was wrong.
>
> > There is an explicit claim that ungrokking differs from CF because there is “only bad test loss.”
>
> Thanks for pointing this out, we agree this is incorrect and will remove it. (We don't think this changes the overall point.)

---

### Official Review · Reviewer_Rk5Z · 2023-11-04

**Soundness:** 3 good
**Presentation:** 3 good
**Contribution:** 2 fair
**Rating:** 5
**Confidence:** 4

**Summary:**

This paper studies grokking via the lens of circuit efficiency. In particular, they conjecture the existence of both memorization and generalization circuits, which have different efficiency, measured by parameter norm when given the same predictive performance. Based on their analysis, they also predict theoretically and verify empirically two new phenomenon called ungrokking and semi-grokking.

**Strengths:**

* The paper is well written and easy to follow.
* The story is in general sound and nicely supported by empirical results
* Enrich the literature of grokking by discovering semi-grokking and un-grokking

**Weaknesses:**

* Although I find the general story to be believable, some details are either incomplete or could have alternative explanations. See the question part.

**Questions:**

* I'm not sure about the terminology "circuit efficiency", since there is no mechanistic interpretability literally picking out circuits.
* I would like to see more analysis on semi-grokking. For example, what are these semi-generalized algorithms doing? For modular addition, one may expect some of semi-generalizing algorithms somehow learn the symmetry of two inputs (Abelian), but fail to learn the more sophisticated generalization patterns.
* For ungrokking, is there a theory for predicting the phase transition point? It would be nice to have a theory (at least some analysis) regarding the critical data size.
* Also for ungrokking, the explanation in the paper is that when the dataset size is small, the memorizing circuit is more efficient than the generalization circuit. Maybe I missed something but I didn't see empirical evidence for that. An alternative explanation could be: memorization and generalization circuits are equally efficient, so a circuit basically randomly wanders around, but they are more memorization circuits than generalization circuits. As a result, the network is more likely to end up being a memorization circuit just because there are more memorization circuits, but not because they are more efficient. In general, maybe both factors are contributing. My point is: there could be alternative hypotheses for ungrokking, and the authors seem overly confident with their claims with limited evidence.

---

> ### Author Response · Authors · 2023-11-19
>
> Thanks for the review! We’re glad that you found our explanation to be sound and supported by empirical results, and that you appreciated the discovery of semi-grokking and ungrokking. We respond to your questions below.
>
> > I'm not sure about the terminology "circuit efficiency", since there is no mechanistic interpretability literally picking out circuits.
>
> We are relying here on Nanda et al (2023) – they perform mechanistic interpretability on the modular addition task and pick out the generalising circuit. We have a short primer on their results in Appendix C, though the full description of their results is of course in their paper.
>
> > I would like to see more analysis on semi-grokking. For example, what are these semi-generalized algorithms doing? For modular addition, one may expect some of semi-generalizing algorithms somehow learn the symmetry of two inputs (Abelian), but fail to learn the more sophisticated generalization patterns.
>
> Our theory predicts that the semi-generalizing network is simply running both the generalising and memorising algorithms simultaneously, and combining their outputs when making its prediction. In fact it is precisely because our theory predicted this that we knew to look for semi-grokking – we did not randomly stumble upon it (it is quite a rare behaviour). So we’re fairly confident that this is what’s happening. The reason for partial test accuracy is because both circuits have noisy outputs, and so when they are evenly matched the generalising circuit produces higher logits about 50% of the time.
>
> We wanted to more clearly show that this was happening by creating metrics that could separate the generalisation and memorisation circuits, and experimented with a variety of proposals for such metrics. Unfortunately, we did not find any metric that we were convinced accurately separated the circuits. An example proposal is given in Appendix C.
>
> It is possible in modular addition to learn the symmetry of two inputs, such that it generalises from a + b to b + a. This does happen in practice. However, we can easily rule this possibility out by considering the test accuracy: if you train on X% of the full dataset, then the test accuracy due to symmetry will be around X% (given a test input x + y, there is an X% chance that we trained on y + x). In semi-grokking, networks frequently have very different test accuracies (see Figure 5), showing that it is not just learning the symmetry heuristic.
>
> We do see other interesting and unexplained behaviour in semi-grokking, for example, the existence of multiple plateaus. A simple hypothesis we offer is the existence of multiple generalising circuits that the network transitions between, but ultimately left this for future work.
>
> > For ungrokking, is there a theory for predicting the phase transition point? It would be nice to have a theory (at least some analysis) regarding the critical data size.
>
> Ultimately $D_{\text{crit}}$ is determined by the efficiencies of the two circuits. It is very hard to determine the efficiency of a circuit completely theoretically, since efficiency depends on how the algorithm is implemented in the neural network, which we usually do not know in advance. However, it is often possible to determine it via measurement: in particular, we can measure the efficiencies of memorisation and generalisation and see where the crossover happens. Indeed, from Figure 3a and 3c, we can see that the crossover point should be around the point where memorization has param norm ~31, which corresponds to a dataset size of just over 500. This is then confirmed in Figure 4.
>
> > Also for ungrokking, the explanation in the paper is that when the dataset size is small, the memorizing circuit is more efficient than the generalization circuit. Maybe I missed something but I didn't see empirical evidence for that.
>
> We do have empirical evidence here – Figure 3a shows that at dataset sizes of ~500, memorisation has parameter norms under 30 (sometimes hitting 25), whereas in Figure 3c we see that generalisation always has parameter norms above 28 (and is roughly independent of dataset size).

---

> > ### Comment · Reviewer_Rk5Z · 2023-11-21
> > **Thanks for the response**
> >
> > * "We do have empirical evidence here – Figure 3a shows that at dataset sizes of ~500, memorisation has parameter norms under 30 (sometimes hitting 25), whereas in Figure 3c we see that generalisation always has parameter norms above 28 (and is roughly independent of dataset size)."
> >
> > Thanks for pointing this out! I indeed missed the point the first time I read it.
> >
> > * "However, we can easily rule this possibility out by considering the test accuracy: if you train on X% of the full dataset, then the test accuracy due to symmetry will be around X%"
> >
> > Things might be slightly more complicated than this estimation. For example, if both a + b and b + a are in the training dataset, there's no extra generalization to test set.
> >
> > I would like to thank the authors for their response. I will keep my score. This paper is a reasonable narrative of grokking, but I personally do not find it transformatively novel. I'd appreciate more efforts on characterizing ungrokking and semigrokking in more details.

---

> ### Author Response · Authors · 2023-11-22
> **Clarifying the point on test accuracy**
>
> > Things might be slightly more complicated than this estimation. For example, if both a + b and b + a are in the training dataset, there's no extra generalization to test set.
>
> It still works out as we described.
>
> Suppose that the training dataset fraction is $p$, that is, a randomly selected input would have probability $p$ of being in the training set. Then for some arbitrary $a$ and $b$, let us consider $x_1$ to be the input "$a + b$" and $x_2$ to be "$b + a$".
>
> There are four possibilities:
>
> 1. $x_1 \in D_{\text{train}}$ and $x_2 \in D_{\text{train}}$: This happens with probability $p^2$. Neither example is in the test set.
>
> 2. $x_1 \in D_{\text{train}}$ and $x_2 \in D_{\text{test}}$: This happens with probability $p(1-p)$. $x_2$ is in the test set and is answered correctly using symmetry.
>
> 3. $x_1 \in D_{\text{test}}$ and $x_2 \in D_{\text{train}}$: This happens with probability $p(1-p)$. $x_1$ is in the test set and is answered correctly using symmetry.
>
> 4. $x_1 \in D_{\text{test}}$ and $x_2 \in D_{\text{test}}$: This happens with probability $(1-p)(1-p)$. Both $x_1$ and $x_2$ are in the test set, and both are answered at random chance (which we'll round to zero accuracy for simplicity).
>
> Then the overall test accuracy becomes the ratio of cases 2 + 3 (successful test answers) to cases 2 + 3 + 4 (all test answers):
>
> $\text{Test accuracy} = \frac{2p(1-p)}{2p(1-p) + 2(1-p)(1-p)} = \frac{2p(1-p)}{2(1-p)} = p$, as we claimed above.

---

### Official Review · Reviewer_A2EW · 2023-11-05

**Soundness:** 3 good
**Presentation:** 3 good
**Contribution:** 3 good
**Rating:** 6
**Confidence:** 4

**Summary:**

This paper explains the grokking phenomenon through the so-called "circuit efficiency," where a circuit means a neural net with certain weights, and the efficiency refers to the parameter norm of a circuit that achieves a small training loss. This paper points out three ingredients in combination that can cause grokking: (1) the existence of a circuit that generalizes well and another circuit that doesn't; (2) the generalizing circuit is more "efficient"; (3) the training algorithm needs to take a long time to find the generalizing circuit.

This implies that the dataset size may be important: when it is smaller than a critical value, the memorizing circuit could be more efficient; when it roughly equals the critical value, the final test accuracy may not be close to 100%.

**Strengths:**

1. The grokking phenomenon being studied in this paper is very puzzling and important.
2. This paper provides simple and intuitive arguments that can partially explain grokking.
3. The importance of the three ingredients and dataset size is validated by experiments.
4. The explanation provided in the paper also leads to the discovery of the "ungrokking" and "semi-grokking" phenomena.

**Weaknesses:**

1. Although the paper claims that they provide a "theory" for grokking, there are no real theorems in the main paper. Many key concepts, such as circuit efficiency, are not defined with formal math, either. I encourage the authors to spend more effort to formulate and present their intuitive arguments with rigorous math.
2. Although the explanation provided by the paper seems intuitive, several key puzzles are still left unexplained, even if we follow the authors' argument with 3 ingredients. This paper mainly explains the second ingredient ("the generalizing circuit is more efficient"). However, the more interesting ones are the first and third ingredients, which may have a closer relation to practice but the paper doesn't make much progress on them.
    * (Related to the first ingredient.) Why are there only two circuits instead of a series of circuits that continuously trade-off between efficiency and training speed? Understanding this is crucial to understand why the transition in grokking is very sharp.
    * (Related to the third ingredient.) Why does the generalizing circuit need more time to be learned? If we just search circuits among those with small parameter norms (or being efficient under other efficiency measures), can we completely avoid meeting memorizing circuits during training? Understanding this is crucial to making neural nets learn algebra/reasoning/algorithmic tasks more efficiently in time.
3. The newly discovered phenomena in the paper, ungrokking and semi-grokking, are indeed very interesting, but a minor weakness is that it is a bit unclear to me whether these phenomena will be of much practical use in the near future.

**Questions:**

This paper is of good quality overall. My main concerns are the weaknesses 1 and 2 above. I would like to know if the authors would like to (1) make the arguments more formal; (2) point out some useful insights about the first and third ingredients that I could have missed.


================

Post-rebuttal update:

Thank the authors for their response. This paper is of good quality overall, but I still feel that the lack of rigor remains a weakness with this theory paper, especially when explaining the first and third ingredients. I would like to keep my score.

---

> ### Author Response · Authors · 2023-11-19
>
> Thanks for the review! We’re happy to see that you think we explain an important and puzzling phenomenon, and appreciate the validation of our theory in its prediction of two novel phenomena.
>
> ## Theory
>
> We actually do have rigorous maths underpinning our theory (Appendix B5, Appendix D). However, this maths is more in the sense of an economist’s mathematical model, than theorems about neural networks.
>
> We are taking an approach more similar to that of the natural sciences, which have extensive experience with observing, understanding, and explaining confusing phenomena. For example, upon Googling “theory”, we find the following top results:
>
> 1. “a supposition or a system of ideas intended to explain something, especially one based on general principles independent of the thing to be explained”
> 2. “a theory not only explains known facts; it also allows scientists to make predictions of what they should observe if a theory is true”
>
> We think our contributions fit these definitions. We agree that a theory gains significant support to the extent that it can be formalised with rigorous maths – but we do so by constructing mathematical models that abstract away details of real systems. In particular, in our mathematical model in Appendix B.5:
>
> - We assume that the two circuits are clearly separable from each other
> - Their “strength” can be represented by parameters $w_M$ and $w_G$
> - Their “efficiency” can be represented by parameters $P_M$ and $P_G$, each meant to quantify the parameter norm of the circuit for given logits
> - Their “speed of learning” can be controlled by splitting $w_M = w_{M1} w_{M2}$ and $w_G = w_{G1} w_{G2}$ and learning $w_{M1}, w_{M2}, w_{G1}, w_{G2}$ with gradient descent.
>
> Given this fully formalised and rigorous mathematical model, we:
> - Prove a theorem (Theorem D.4) that the weights that minimise loss will favour efficient circuits (we also derive a quantitative relationship between circuit strength and circuit efficiency).
> - Demonstrate with simulations (Figure 2) that grokking arises when the three ingredients are present, and doesn’t arise if either of the latter two ingredients are not present.
>
> Based on feedback on an earlier version of this paper, we moved most of this work to the appendices, and only kept Figure 2 (part of the results) in the main body of the paper. We would be interested in suggestions on how to better present the maths in the main paper given our space constraints.
>
> ## Insights on the first and third ingredients
>
> **First ingredient:** We discuss this briefly in the Discussion section, but we can elaborate on it. Consider a more typical task, such as MNIST. There are a wide variety of simple heuristics such as edges and curves that provide some generalisation. These heuristics can all separately contribute to test performance, independently of other heuristics, and will have a wide variety of efficiencies and learning speeds, as the reviewer suggests.
>
> By contrast, in modular addition it is not clear what heuristics the model could learn that provide partial generalisation. (One exception: models can learn a commutative lookup table – we in fact observed this in some of our runs.) It turns out that this applies for many algorithmic, symbolic tasks, which is what produces grokking.
>
> Inspired by your question, we conducted experiments on MNIST, which generally validated the theoretical picture painted above (see Figures 7 and 10 in the updated paper).
>
> **Third ingredient:** We do have a hypothesis about why the generalizing circuit is learned more slowly, inspired by Jermyn & Shlegeris (2022). A generalizing circuit for an algorithmic problem is likely to be composed of many sequential operations / subcircuits, such that all of the operations have to be performed in order to achieve any reduction in loss. Since the gradient towards any one component will depend on the strength of all the other components, this causes slow, then fast learning. This inspires our toy example, illustrated in Figure 2 and described in mathematical detail in Appendix B.5. However, we still view this hypothesis as speculative, so we did not highlight it in the main body of the paper.
>
> **Avoiding memorising circuits during training:** We tried a few experiments that didn't make it into the paper because we didn't have time to investigate them conclusively.
> 1. We tested specifically the multiple components hypothesis by initialising the embeddings to the final learned embeddings, while using random initialisation for the rest of the network, and indeed found that the network generalised immediately without memorising, presumably by removing the need to learn one of the components.
> 2. We also investigated lottery tickets, and initialising with the winning lottery ticket (around 5% of the full network) saw immediate generalisation, possibly by making the memorisation circuit slower to learn while leaving the generalising circuit unaffected.

---

### Meta-Review · Area_Chair_x3CB · 2023-12-11

**Metareview:**

This submission investigates the phenomenon of grokking in neural networks, proposing a theory based on the concept of 'circuit efficiency' to explain how neural networks transition from memorizing to generalizing solutions. The authors provide empirical evidence to support their theory, including the introduction of novel phenomena such as 'ungrokking' and 'semi-grokking.'

Reviewers brought up major issues. For example, key concepts, like 'circuit efficiency,' lack clear and thorough definitions. Further, the novelty of the paper's contributions is not convincingly differentiated from existing literature.

**Justification For Why Not Higher Score:**

The reviewer support is not strong enough for acceptance.

**Justification For Why Not Lower Score:**

N/A

---

### Decision · Program_Chairs · 2024-01-16

Reject